# MIND: Multi-rationale INtegrated Discriminative Reasoning Framework for Multi-modal Large Models

Chuang Yu [1 2 3]   Jinmiao Zhao [1 2]   Mingxuan Zhao [4]   Yunpeng Liu [1 †]   Xiujun Shu [2]
Yuanhao Feng [5]   Bo Wang [6]   Xiangyu Yue [3 †]

## Abstract

Recently, multimodal large language models (MLLMs) have been widely applied to reasoning tasks. However, they suffer from limited multi-rationale semantic modeling, insufficient logical robustness, and susceptibility to misleading cues. Therefore, we propose a **M**ulti-rationale **IN**tegrated **D**iscriminative (**MIND**) reasoning framework, which is designed to endow MLLMs with human-like cognitive abilities of "Understand → Rethink → Correct", and achieves a paradigm evolution from passive imitation-based reasoning to active discriminative reasoning. Specifically, we introduce a Rationale Augmentation and Discrimination (RAD) paradigm, which provides a unified and extensible data foundation. Meanwhile, we design a Progressive Two-stage Correction Learning (P2CL) strategy. The first phase enhances multi-rationale positive learning, while the second phase enables active logic discrimination and correction. In addition, to mitigate representation entanglement in the multi-rationale semantic space, we propose a Multi-rationale Contrastive Alignment (MCA) optimization strategy. Extensive experiments show that our MIND achieves SOTA performance on multiple public datasets. Our data and code are available at https://github.com/YuChuang1205/MIND.

## 1. Introduction

With the rapid advancement of multimodal large language models (MLLMs) across tasks such as visual question an-

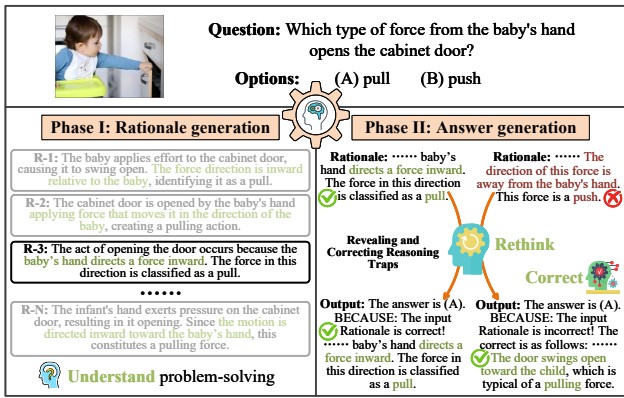

*Figure 1.* Illustration of MIND's "Understand → Rethink → Correct" paradigm. It includes two phases: Rationale (reasoning chain) generation and Answer generation. In **Phase I**, the model focuses on understanding key problem-solving elements. In **Phase II**, the model rethinks rationales and corrects flawed reasoning.

swering (VQA) (Cheng et al., 2025; Yang et al., 2025c; Tu et al., 2025; Chen et al., 2024), visual reasoning (Dong et al., 2025; Yang et al., 2025b; Wang et al., 2024a), and cross-modal understanding (Jain et al., 2024; Huang et al., 2025; Wu et al., 2024a), reasoning ability has emerged as a key indicator of a model's intelligence. To enhance this, recent research has proposed Multimodal Chain-of-Thought (MCoT) (Zhang et al., 2024b), which incorporates intermediate reasoning steps. This allows models to generate reasoning chains in a step-by-step manner, thereby improving logical transparency (Wang et al., 2025). However, existing MCoT methods (Zhang et al., 2024b; Tan et al., 2024a; He et al., 2024; Wang et al., 2024b) still predominantly rely on single-rationale supervision during training. This paradigm tends to drive models to learn only the surface-level mapping toward a standard answer, failing to capture the diversity and complexity inherent in human reasoning. Consequently, when confronted with ambiguous, incorrect, or misleading explanations, these models often exhibit rigid reasoning patterns, weak logical robustness, and limited reasoning discrimination or self-correction capabilities. In contrast, human reasoning is not a static, linear process but rather a dynamic cycle involving diverse rationales, reflective falsification, and self-correction (Yax et al., 2024).

[†]Corresponding authors [1]Shenyang Institute of Automation, Chinese Academy of Sciences [2]University of Chinese Academy of Sciences [3]MMLab, CUHK [4]HKUST(GZ) [5]Peking University [6]NWPU. Correspondence to: Yunpeng Liu <ypliu@sia.cn>, Xiangyu Yue <xyyue@ie.cuhk.edu.hk>.

*Proceedings of the 43rd International Conference on Machine Learning*, Seoul, South Korea. PMLR 306, 2026. Copyright 2026 by the author(s).

Current MLLMs generally lack the ability to model and discriminate among diverse rationale relations, making them vulnerable to misleading or adversarial information in complex scenarios. We believe that relying solely on single-rationale supervision fails to capture the diversity and self-corrective nature of human reasoning. As shown in Figure 1, MLLMs need to learn both the semantic consistency of correct reasoning and the discriminative boundaries of incorrect reasoning within a multi-rationale space, and further develop self-correcting capabilities, thereby facilitating a paradigm evolution from passive imitation-based reasoning to active discriminative reasoning. Therefore, endowing MLLMs with the capability to understand multiple rationales, identify logical inconsistencies, and perform self-correction is crucial for advancing intelligent reasoning from imitation to true cognitive intelligence.

Based on the above motivation, we propose the first Multi-rationale INtegrated Discriminative (MIND) reasoning framework for MLLMs. Unlike traditional single-rationale supervision, MIND incorporates diverse positive rationales to model the diversity of human reasoning, while simultaneously leveraging challenging negative rationales to reveal potential reasoning pitfalls. Specifically, we propose a Rationale Augmentation and Discrimination (RAD) paradigm that automatically and efficiently generates diverse rationales. Unlike conventional datasets that provide only a single rationale, RAD adopts a positive/negative co-generation mechanism to explicitly model diverse reasoning and potential reasoning traps, providing a unified and extensible data foundation. Meanwhile, inspired by the human cognitive process of "Understand → Rethink → Correct", we design a Progressive Two-stage Correction Learning (P2CL) strategy to drive the evolution from passive imitation-based reasoning to active discriminative reasoning. In Phase I, diverse positive rationales are used to enhance semantic understanding and multi-rationale logical modeling, while Phase II leverages both positive and negative rationales to guide the model in identifying and correcting erroneous reasoning, forming a self-reflective and logic-repairing reasoning mechanism. Additionally, to further improve the model's discriminative ability within the multi-rationale semantic space, we propose a Multi-rationale Contrastive Alignment (MCA) optimization strategy. By aggregating hard positive rationales and separating hard negative rationales in the embedding space, MCA adaptively expands semantic boundaries, significantly enhancing the model's logical consistency and discrimination sensitivity. The contributions can be summarized as follows:

- We propose the first MIND reasoning framework for MLLMs. By introducing diverse positive rationales to model the diversity of human reasoning and incorporating challenging negative rationales to reveal and correct potential reasoning pitfalls, MIND drives a paradigm

evolution from passive imitation-based reasoning to active discriminative reasoning.

- We develop a RAD paradigm that automatically and efficiently generates diverse positive rationales and semantically inverted challenging negative rationales, providing a unified and extensible data foundation.

- Inspired by the human cognitive mechanism, we design a P2CL strategy. The first phase enhances multi-rationale positive learning, while the second phase enables active logic discrimination and correction.

- To further enhance discrimination, we construct a MCA optimization strategy that achieves effective positive rationale aggregation and semantic conflict separation.

## 2. Related Work

**Prompt-based MCoT Reasoning** aims to enhance multimodal reasoning ability by explicitly inducing models to generate step-by-step reasoning through carefully designed prompts. Typical methods adopt prompts such as "Let's think step by step!" to encourage sequential reasoning, while employing explicit task planning or fixed reasoning templates to constrain logical paths (Singh et al., 2023; Chen et al., 2023; Luo et al., 2024; Wu et al., 2023b; Mitra et al., 2024; Zhang et al., 2024a; Wang et al., 2024d; Zhao et al., 2023). To further strengthen visual-semantic understanding, some studies incorporate external tools (Wu et al., 2024c; Gao et al., 2024; Liu et al., 2023; Tan et al., 2024b) or transform visual features into textual form (Meng et al., 2023; Hu et al., 2024; Wu et al., 2024b), allowing language models to better fuse cross-modal information. Prompt-based methods require no additional training and are highly flexible, making them well-suited for resource-constrained tasks. However, they are prompt-sensitive and lack error discrimination and correction mechanisms, which makes them prone to logical drift or hallucination (Wang et al., 2025).

**Learning-based MCoT Reasoning** explicitly models the reasoning generation during training, enabling models to internalize reasoning abilities through supervised learning. As an early milestone, Multimodal-CoT (Zhang et al., 2024b) established a two-stage learning framework that paved the way for later studies (Zheng et al., 2023; Tan et al., 2024a; He et al., 2024; Wang et al., 2024b). Subsequently, MC-CoT (Tan et al., 2024a) enhances the logical robustness by introducing multimodal consistency constraints and majority-voting mechanisms. T-SciQ (Wang et al., 2024b) leverages high-quality rationales generated by powerful language models for cross-modal reasoning transfer. To further mitigate hallucination, some works incorporate diffusion or retrieval-based auxiliary tools (Dong et al., 2025; Wu et al., 2024b; Meng et al., 2023; Zhong et al., 2024; Lee

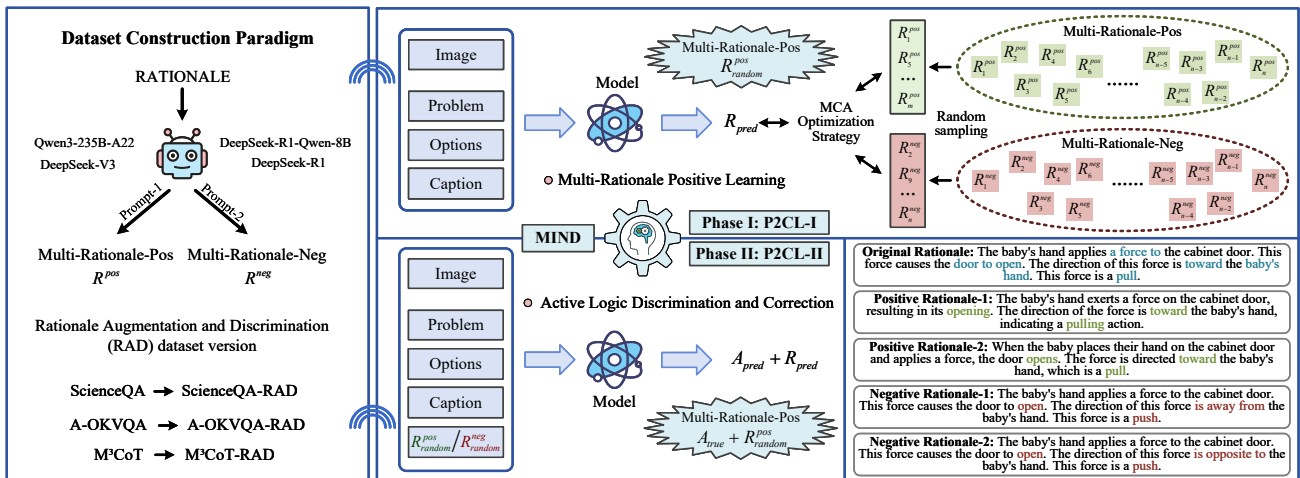

Figure 2. Overview of the MIND reasoning framework. The blue circle denotes the format of the supervision signal.

et al., 2024; Li et al., 2025; Koh et al., 2023). For example, VoT (Wu et al., 2024b) enables text-only models to imagine visual reasoning paths. Overall, learning-based methods achieve stronger logical consistency and semantic stability than prompt-based methods. However, they rely heavily on single-rationale supervision and lack mechanisms for identifying and correcting flawed reasoning. To address this, we aim to endow MLLMs with human-like cognitive abilities of "Understand → Rethink → Correct", achieving a paradigm evolution from passive imitation-based reasoning to active discriminative reasoning.

## 3. Methods

### 3.1. Overview

Inspired by human cognitive processes, we propose the MIND reasoning framework. As shown in Figure 2, MIND consists of two key components: the RAD dataset construction paradigm and the MIND learning architecture. For the data construction, the RAD paradigm provides MIND with multi-rationale training samples. Using well-designed positive and negative prompts, existing large models are guided to generate diverse positive rationales and challenging negative rationales under controlled conditions. For model learning, the MIND aims to enhance human-like cognitive ability. Unlike conventional methods that rely on single-rationale supervision, MIND incorporates diverse positive and negative rationales during training and employs a P2CL strategy to achieve alignment with the human cognitive pattern of "Understand → Rethink → Correct". Specifically, Phase I (P2CL-I) focuses on multi-rationale positive learning. This phase encourages the MLLMs to grasp the essence of problem-solving (understanding) rather than direct mapping relationships (rote memorization). Phase II (P2CL-II) emphasizes active logic discrimination and correction, using both positive and negative rationales to identify and

correct reasoning errors. In addition, the MIND incorporates a MCA optimization strategy, which enforces semantic aggregation across consistent rationales and separation of conflicting semantics. In summary, the proposed MIND, through the synergistic interaction of RAD, P2CL and MCA, achieves a transition from shallow imitative reasoning to deep cognitive reasoning.

### 3.2. RAD Paradigm

To generate diverse rationales, we propose a RAD dataset construction paradigm, which systematically generates diverse positive rationales and semantically inverted negative rationales to model the diversity and discriminative nature of human reasoning. Unlike conventional VQA datasets (Chen et al., 2024; Lu et al., 2022; Schwenk et al., 2022) that provide a single standard rationale, RAD offers a unified and extensible data foundation.

From Figure 3, for each sample $S = \{I, Q, O, C, A, R_{gt}\}$ consisting of an image $I$, question $Q$, options $O$, image caption $C$, answer $A$, and the original rationale $R_{gt}$, we design a structured prompt template to guide existing large models to automatically and efficiently generate diverse rationales. To mitigate the high computational and time costs of generating rationales one by one, RAD adopts a batch-wise generation mechanism in its prompt design, allowing multiple rationales to be generated in a single interaction, which significantly improves data construction efficiency. In addition, to ensure that the generated rationales meet semantic requirements, we incorporate task-related information, such as the question, options, and answer, into the input to help large models better understand the reasoning context and semantic constraints, thereby avoiding the generation of rationales that do not meet the requirements. Specifically, we design two types of prompts: *Positive Prompts* and *Negative Prompts*. Details are provided in the Appendix A.

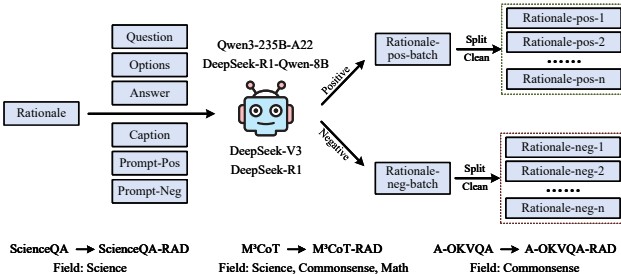

*Figure 3.* Overview of the RAD paradigm.

Under the guidance of the positive and negative prompts, the large model generates multiple candidate rationales for each sample. We then perform rationale splitting and cleaning to ensure that the generated content adheres to both format and semantic standards. Specifically, the model output is first split using the predefined delimiter, separating the batch-generated rationales into individual samples. Next, each rationale undergoes filtering and cleaning to ensure textual conciseness and consistency. All valid positive and negative rationales are stored separately in *Multi-Rationale-Pos* and *Multi-Rationale-Neg* pools, forming the final multi-rationale augmented sample:

$$S_{RAD} = \{I, Q, O, C, A, R_{gt}, \{R_i^+\}, \{R_j^-\}\}, \quad (1)$$

where $\{R_i^+\}$ and $\{R_j^-\}$ denote the set of positive rationales and negative rationales. This collaborative design of positive and negative prompts enables the construction of high-quality and well-structured training samples for subsequent MIND learning.

### 3.3. P2CL Strategy

Inspired by the human cognitive patterns of "Understand $\rightarrow$ Rethink $\rightarrow$ Correct", we design a P2CL strategy. This strategy divides the training process into two complementary phases: multi-rationale positive learning (P2CL-I) and active logic discrimination and correction (P2CL-II).

In P2CL-I, the model uses random sampling in the *Multi-Rationale-Pos* pool as supervision, aiming to learn the shared causal and logical chains expressed in different linguistic forms, thereby avoiding overfitting to a single standard rationale. Given a multimodal sample $S_{RAD} = \{I, Q, O, C, R_{gt}, \{R_i^+\}_{i=1}^{M}\}$, the model jointly encodes visual and textual inputs via the multimodal encoder and a language decoder to generate the rationale $\hat{R}$. We employ a maximum likelihood generation objective to encourage the model to capture consistent logic across diverse rationales:

$$\mathcal{L}_{\text{pos}} = -\mathbb{E}_{R^+} \sum_t \log p(\hat{R}_t = R_t^+ \mid I, Q, O, C). \quad (2)$$

To further enhance the semantic-boundary discrimination and embedding consistency, P2CL-I is jointly optimized

with the MCA optimization strategy (please see Section 3.4), whose loss term is denoted as $\mathcal{L}_{\text{mca}}$. Thus, the total optimization objective of P2CL-I is as follows:

$$\mathcal{L}_{P-I} = \mathcal{L}_{\text{pos}} + \alpha \cdot \mathcal{L}_{\text{mca}} \quad (3)$$

In P2CL-II, the model uses positive-negative rationale pairs $R_{cond} \in \{R_i^+, R_j^-\}$ as semantic cues, and explicitly models error identification and logical correction through a unified joint generation objective for both answers and rationales. Given a multimodal sample input $Q' = \{I, Q, O, C, R_{\text{cond}}\}$, the model takes either a positive or negative rationale as input and generates a concatenated output sequence of the answer and rationale $[\hat{A}, \hat{R}^+]$:

$$\mathcal{L}_{P-II} = -\mathbb{E}_{R_{\text{cond}}} \sum_t \log p([\hat{A}, \hat{R}_t^+] = [A, R_t^+] \mid Q'), \quad (4)$$

where $A$ and $R_t^+$ denote the correct answer and the target rationale, respectively. When the input rationale is positive, the model is constrained to maintain expression stability. When the input rationale is negative, the model is required to identify logical deviations in the rationale and correct them. P2CL-II enhances the model's discriminative reasoning and logical correction capabilities.

After the two-stage training, the model learns semantic consistency, discriminative boundaries, and self-corrective capability. The P2CL strategy drives a paradigm evolution from passive imitation-based reasoning to active discriminative reasoning, empowering the model with human-like cognitive capabilities.

### 3.4. MCA Optimization Strategy

Although the P2CL strategy progressively establishes semantic consistency and discriminative reasoning ability, representation instability may still occur within the multi-rationale semantic space. To address this issue, we further propose a MCA optimization strategy, which explicitly enforces the semantic aggregation of correct rationales and separation of incorrect rationales in the embedding space through a hard-rationale mining mechanism and contrastive semantic constraints.

First, the multimodal encoder and language decoder jointly model the multimodal inputs to obtain the predicted embedding, which is then projected into a unified semantic contrastive space via a linear mapping layer $g_\phi(\cdot)$. For each sample, the sets of positive and negative rationales are denoted as $\{R_i^+\}$ and $\{R_j^-\}$, respectively. After encoding, their embeddings can be represented as:

$$h^{+/-} = g_\phi\Big(f_\theta(I, Q, O, C, R^{+/-})\Big). \quad (5)$$

Second, we randomly sample $N$ rationales from both positive and negative sets and compute their cosine similarities

with predicted embedding in each iteration:

$$s_i^+ = \text{sim}(h_{\text{pred}}, h_i^+), \quad s_j^- = \text{sim}(h_{\text{pred}}, h_j^-). \quad (6)$$

Third, to focus effective parameter updates guided by the most discriminative rationales, the MCA strategy performs dual hard rationale mining for both positive and negative sets. For positive rationales $\{s_i^+\}$, the Bottom-k (with the lowest similarity) is selected. For negative rationales $\{s_j^-\}$, the Top-k (with the highest similarity) is selected:

$$S_{\text{hard}}^+ = \text{Bottom-}k(\{s_i^+\}), \quad S_{\text{hard}}^- = Top\text{-}k(\{s_j^-\}). \quad (7)$$

Finally, to explicitly enlarge the distance between positive and negative rationales in the embedding space, we define a margin-based contrastive loss:

$$\mathcal{L}_{\text{con}} = \text{ReLU}(\bar{S}_{\text{hard}}^- + m - \bar{S}_{\text{hard}}^+), \quad (8)$$

where $m$ is the margin. $\bar{S}_{\text{hard}}^-$ and $\bar{S}_{\text{hard}}^+$ denote the mean cosine similarities of the negative Top-k and positive Bottom-k rationales, respectively. This loss encourages the model to pull closer the hard positive rationales and push away the hard negative rationales, thereby forming clear semantic boundaries in the embedding space.

Directly using contrast constraints may prematurely force the aggregation or separation of rationales before the semantic space has fully converged, potentially increasing the risk of semantic collapse and representation drift. Therefore, the MCA strategy is coupled with P2CL-I to achieve synergistic optimization between generative supervision and contrastive alignment in our experiments. This mechanism suppresses semantic divergence in early training and reinforces logical consistency in later stages, ultimately enabling the model to form a stable semantic space.

## 4. Experiments

### 4.1. Datasets

We conduct systematic experiments on three VQA datasets covering different domains, namely, ScienceQA (Lu et al., 2022), A-OKVQA (Schwenk et al., 2022), and M³CoT (Chen et al., 2024). ScienceQA focuses on science and contains 21,208 samples. A-OKVQA targets commonsense question answering and includes 24,903 samples. M³CoT integrates science, commonsense, and mathematics, comprising 11,328 samples. For all datasets, the data partitioning remains consistent with their original versions to ensure fair comparison. Meanwhile, based on the proposed RAD paradigm, we build corresponding RAD-extended versions for each dataset, namely ScienceQA-RAD, A-OKVQA-RAD, and M³CoT-RAD. Compared with original datasets, the rationales in these RAD versions are expanded by factors of 1000×, 1000×, and 500×, respectively.

### 4.2. Implementation Details

Following previous research settings, we adopt a T5-based encoder-decoder architecture (Raffel et al., 2020) and explore two model scales: Base (223M) and Large (738M). The model is initialized with FLAN-Alpaca weights (Chung et al., 2024). Visual features are extracted using a frozen BLIP2-flan-t5-xxl (Li et al., 2023) encoder, while image captions are generated by frozen Qwen2.5-VL-72B (Bai et al., 2025). The learning rate and batch size are set to $8e^{-5}$ and 8. The hyperparameters $m$ and $\alpha$ are set to 0.2 and 1. The maximum input sequence length is 512. We use a fixed random seed of 42 during training. Considering the varying complexity of different datasets, the number of training epochs is set to 200 for ScienceQA-RAD, and 400 for A-OKVQA-RAD and M³CoT-RAD. All experiments are conducted on eight 96GB NVIDIA H20 GPUs.

### 4.3. Comparison with SOTA Methods

We conduct systematic comparisons with a variety of excellent methods. Since MIND is developed upon Multimodal-CoT, we particularly highlight the performance comparison with it to show the performance improvements. More visualizations can be found in the Appendix C.

*1) Evaluation on the ScienceQA Dataset.* As shown in Table 1, the $\text{MIND}_{base}$ model achieves an average accuracy of 92.29%, reaching the current SOTA performance. First, compared to early VQA and cross-modal understanding models (e.g., MCAN (Yu et al., 2019)), $\text{MIND}_{base}$ achieves approximately 30% improvement. Second, compared to large language models such as GPT-3.5 (Lu et al., 2022), $\text{MIND}_{base}$ outperforms them by 0.54% - 18.32%. Third, for larger MLLMs (such as LLaVA (Liu et al., 2024b)), $\text{MIND}_{base}$ achieves 1.37% - 7.10% improvement with only 223M parameters. Finally, compared with recent multimodal CoT-based methods, our $\text{MIND}_{base}$ achieves superior performance with the same or fewer parameters. Compared to MC-CoT$_{base}$ (Tan et al., 2024a), DPMM-CoT$_{base}$ (He et al., 2024), and Multimodal-T-SciQ$_{base}$ (Wang et al., 2024b), $\text{MIND}_{base}$ improves performance by 1.65% (from 90.64% to 92.29%), 1.32% (from 90.97% to 92.29%), and 0.32% (from 91.97% to 92.29%), respectively. Notably, compared to Multimodal-CoT (Zhang et al., 2024b), $\text{MIND}_{base}$ improves 6.98%.

*2) Evaluation on the A-OKVQA Dataset.* Compared to ScienceQA, A-OKVQA has a stronger knowledge dependency and openness. From Table 2, $\text{MIND}_{base}$ achieves an accuracy of 70.6%, reaching the current SOTA performance. First, compared to few-shot CoT-based methods (CoT (Wei et al., 2022), Pica (Yang et al., 2022), ClipCap (Mokady et al., 2021), and IPVR (Chen et al., 2023)), $\text{MIND}_{base}$ achieves an accuracy improvement of 11.9% - 25.5%. Second, compared to the vision-language fine-tuning methods

*Table 1.* Main results (%) on the ScienceQA dataset. Learning = Learning and training methods. Size = backbone model size. NAT = natural science, SOC = social science, LAN = language science, TXT = text context, IMG = image context, NO = no context, G1-6 = grades 1-6, G7-12 = grades 7-12. Red denotes the best result, and blue denotes the second best result.

| Model | Learning | Size | NAT | SOC | LAN | TXT | IMG | NO | G1-6 | G7-12 | **Avg** |
|---|---|---|---|---|---|---|---|---|---|---|---|
| Random | - | - | 40.28 | 46.13 | 29.25 | 47.45 | 40.08 | 33.66 | 39.35 | 40.67 | 39.83 |
| Human | - | - | 90.23 | 84.97 | 87.48 | 89.60 | 87.50 | 88.10 | 91.59 | 82.42 | 88.40 |
| MCAN (Yu et al., 2019) | Fine-tune | 95M | 56.08 | 46.23 | 58.09 | 59.43 | 51.17 | 55.40 | 51.65 | 59.72 | 54.54 |
| Top-Down (Anderson et al., 2018) | Fine-tune | 70M | 59.50 | 54.33 | 61.82 | 62.90 | 54.88 | 59.79 | 57.27 | 62.16 | 59.02 |
| BAN (Kim et al., 2018) | Fine-tune | 112M | 60.88 | 46.57 | 66.64 | 62.61 | 52.60 | 65.51 | 56.83 | 63.94 | 59.37 |
| DFAF (Gao et al., 2019) | Fine-tune | 74M | 64.03 | 48.82 | 63.55 | 65.88 | 54.49 | 64.11 | 57.12 | 67.17 | 60.72 |
| ViLT (Kim et al., 2021) | Fine-tune | 113M | 60.48 | 63.89 | 60.27 | 63.20 | 61.38 | 57.00 | 60.72 | 61.90 | 61.14 |
| Patch-TRM (Lu et al., 2021) | Fine-tune | 90M | 65.19 | 46.79 | 65.55 | 66.96 | 55.28 | 64.95 | 58.04 | 67.50 | 61.42 |
| VisualBERT (Li et al., 2020) | Fine-tune | 111M | 59.33 | 69.18 | 61.18 | 62.71 | 62.17 | 58.54 | 62.96 | 59.92 | 61.87 |
| GPT-3.5 (Lu et al., 2022) | Few-shot | 175B | 74.64 | 69.74 | 76.00 | 74.44 | 67.28 | 77.42 | 76.80 | 68.89 | 73.97 |
| GPT-3.5 w/ coT (Wei et al., 2022) | Few-shot | 175B | 75.44 | 70.87 | 78.09 | 74.68 | 67.43 | 79.93 | 78.23 | 69.68 | 75.17 |
| ChatGPT w/ coT (Achiam et al., 2023) | Few-shot | 175B | 78.82 | 70.98 | 83.18 | 77.37 | 67.92 | 86.13 | 80.72 | 74.03 | 78.31 |
| GPT-4 w/ coT (Achiam et al., 2023) | Few-shot | - | 85.48 | 72.44 | 90.27 | 82.65 | 71.49 | 92.89 | 86.66 | 79.04 | 83.99 |
| Chameleon (ChatGPT) (Lu et al., 2023) | Few-shot | - | 81.62 | 70.64 | 84.00 | 79.77 | 70.80 | 86.62 | 81.86 | 76.53 | 79.93 |
| Chameleon (GPT-4) (Lu et al., 2023) | Few-shot | - | 89.83 | 74.13 | 89.82 | 88.27 | 77.64 | 92.13 | 88.03 | 83.72 | 86.54 |
| Gemini+RMR (Tan et al., 2024b; Team et al., 2023) | Few-shot | - | 91.79 | 94.26 | 89.64 | 91.40 | 89.69 | 91.01 | 92.84 | 89.78 | 91.75 |
| VaLiK(Qwen2.5-7B) (Liu et al., 2025a) | Zero-shot | 7B | 85.19 | 75.14 | 87.64 | 82.99 | 73.18 | 89.69 | 84.40 | 80.95 | 83.16 |
| VaLiK(Qwen2.5-72B) (Liu et al., 2025a) | Zero-shot | 72B | 85.61 | 75.93 | 90.27 | 84.40 | 74.17 | 92.33 | 85.79 | 82.98 | 84.77 |
| MemVerse(Qwen2.5-7B) (Liu et al., 2025b) | Zero-shot | 7B | 74.51 | 68.50 | 78.73 | 75.92 | 66.19 | 81.95 | 79.70 | 64.73 | 75.62 |
| MemVerse(GPT-4o-mini) (Liu et al., 2025b) | Zero-shot | - | 85.26 | 81.55 | 89.09 | 83.28 | 78.19 | 91.50 | 88.11 | 80.75 | 85.48 |
| LLaMA-Adapter (Zhang et al., 2023) | Fine-tune | 6B | 84.37 | 88.30 | 84.36 | 83.72 | 80.32 | 86.90 | 85.83 | 84.05 | 85.19 |
| LLaVA (Liu et al., 2024b) | Fine-tune | 13B | 90.36 | 95.95 | 88.00 | 89.49 | 88.00 | 90.66 | 90.93 | 90.90 | 90.92 |
| LaVIN (Luo et al., 2023) | Fine-tune | 7B | 89.25 | 94.94 | 85.24 | 88.51 | 87.46 | 88.08 | 90.16 | 88.07 | 89.41 |
| UnifiedQA$_{base}$ (Khashabi et al., 2020) | Fine-tune | 223M | 68.16 | 69.18 | 74.91 | 63.78 | 61.38 | 77.84 | 72.98 | 65.00 | 70.12 |
| UnifiedQA$_{base}$ w/ coT (Lu et al., 2022) | Fine-tune | 223M | 71.00 | 76.04 | 78.91 | 66.42 | 66.53 | 81.81 | 77.06 | 68.82 | 74.11 |
| DDCoT$_{base}$ (Zheng et al., 2023) | Fine-tune | 223M | 88.72 | 86.84 | 84.91 | 87.59 | 83.34 | 88.08 | 88.58 | 85.10 | 87.34 |
| MC-CoT$_{base}$ (Tan et al., 2024a) | Fine-tune | 223M | 91.87 | 84.59 | 93.00 | 92.28 | 88.30 | 92.75 | 90.64 | 90.64 | 90.64 |
| DPMM-CoT$_{base}$ (He et al., 2024) | Fine-tune | 306M | 92.72 | 87.85 | 88.91 | 92.72 | 90.48 | 91.29 | 91.45 | 90.11 | 90.97 |
| Multimodal-T-SciQ$_{base}$ (Wang et al., 2024b) | Fine-tune | 223M | 91.52 | 91.45 | 92.45 | 91.94 | 90.33 | 92.26 | 92.11 | 91.10 | 91.75 |
| Multimodal-CoT$_{base}$ (Zhang et al., 2024b) | Fine-tune | 223M | 84.06 | 92.35 | 82.18 | 82.75 | 82.75 | 84.74 | 85.79 | 84.44 | 85.31 |
| **MIND$_{base}$ (Ours)** | Fine-tune | 223M | 93.07 | 96.74 | 87.09 | 92.42 | 92.76 | 89.27 | 92.91 | 91.17 | 92.29 |
| Improvement | - | - | 9.01 ⇑ | 4.39 ⇑ | 4.91 ⇑ | 9.67 ⇑ | 10.01 ⇑ | 4.53 ⇑ | 7.12 ⇑ | 6.73 ⇑ | 6.98 ⇑ |

*Table 2.* Results (%) on the Multiple-Choice task of A-OKVQA dataset. Red marks the best result, and blue the second best.

| Model | Learning | *Acc.* | Model | Learning | *Acc.* |
|---|---|---|---|---|---|
| CoT | Few-shot | 48.1 | Pythia | Fine-tune | 49.0 |
| Pica | Few-shot | 46.1 | ViLBERT | Fine-tune | 49.1 |
| ClipCap | Few-shot | 56.9 | LXMERT | Fine-tune | 51.4 |
| IPVR (OPT-66B) | Few-shot | 48.6 | KRISP | Fine-tune | 51.9 |
| IPVR (GPT-3) | Few-shot | 58.7 | GPV-2 | Fine-tune | 60.3 |
| Multimodal-CoT$_{base}$ | Fine-tune | 50.6 | **MIND$_{base}$** | Fine-tune | 70.6 |

(Pythia (Jiang et al., 2018), ViLBERT (Lu et al., 2019), LXMERT (Tan & Bansal, 2019), KRISP (Marino et al., 2021), GPV-2 (Kamath et al., 2022), and Multimodal-CoT$_{base}$ (Zhang et al., 2024b)), MIND$_{base}$ outperforms by 10.3% - 21.6%. Finally, compared with Multimodal-CoT$_{base}$, our MIND$_{base}$ achieves a remarkable 20.0% improvement (from 50.6% to 70.6%).

*3) Evaluation on the M³CoT Dataset.* From Table 3, MIND$_{base}$ achieves an average accuracy of 57.38%, outperforming Multimodal-CoT$_{base}$ and MC-CoT$_{base}$ by 12.53% and 3.87%, respectively. Meanwhile, the extended MIND$_{large}$ model surpasses Multimodal-CoT$_{large}$ and MC-CoT$_{large}$ by 12.83% and 3.87%, respectively. In addition, MIND has more prominent advantages compared to other types of models. First, for tool-enhanced models such as HuggingGPT (Shen et al., 2023), MIND achieves signif-

icantly better performance, demonstrating the efficiency of its intrinsic reasoning mechanism. Secondly, for zero-shot multimodal large models, their overall accuracy is only 23.17% - 56.95%. Compared to the optimal GPT-4V, MIND$_{large}$ (738M) achieves a 4.61% improvement at a smaller model size. Finally, for conventional fine-tuning models such as LLaMA-Adapter (Zhang et al., 2023), MIND$_{large}$ achieves 2.06% - 6.67% improvement with fewer parameters.

### 4.4. Ablation Experiments

To fully validate the proposed MIND reasoning framework, we perform detailed ablation experiments. More experiments can be found in the Appendix B.

*1) Break-Down Ablation.* From Table 4, when only the MCA optimization strategy is applied, the model achieves an improvement of 0.07%, indicating that MCA facilitates semantic alignment and feature aggregation, though its standalone impact remains limited. Meanwhile, applying only the P2CL strategy yields a significant gain of 1.86%, demonstrating that multi-rationale positive learning and active logic discrimination and correction play a crucial role in enhancing reasoning. Furthermore, when both strategies are combined, the model performance further rises to 92.29%, achieving a "1 + 1 > 2" synergistic effect. These results

*Table 3.* Main results (%) on the M³CoT dataset. "Random" and "Human" performance are the average accuracy by three attempts. Red denotes the best result, and blue denotes the second best result.

| Model | Size | Science | | | Commonsense | | | Mathematics | | | Total |
|---|---|---|---|---|---|---|---|---|---|---|---|
| | | Lang | Natural | Social | Physical | Social | Temporal | Algebra | Geometry | Theory | |
| Random | - | 32.70 | 30.62 | 26.71 | 32.97 | 22.22 | 20.33 | 35.71 | 27.50 | 23.81 | 28.56 |
| Human | - | 97.83 | 92.62 | 94.31 | 96.28 | 92.41 | 88.71 | 87.23 | 88.75 | 85.71 | 91.61 |
| *Tool-Usage Methods* | | | | | | | | | | | |
| HuggingGPT (Shen et al., 2023) | 175B | 17.57 | 20.93 | 10.33 | 8.70 | 14.75 | 9.76 | 11.35 | 22.50 | 9.52 | 14.60 |
| VisualChatGPT (Wu et al., 2023a) | >175B | 30.09 | 36.28 | 7.78 | 43.48 | 29.92 | 33.33 | 21.99 | 21.25 | 28.57 | 25.92 |
| IdealGPT (You et al., 2023) | - | 31.73 | 31.63 | 26.23 | 56.52 | 50.00 | 26.83 | 20.57 | 30.00 | 38.10 | 32.19 |
| Chameleon (Lu et al., 2023) | - | 43.87 | 26.05 | 25.44 | 39.13 | 37.30 | 48.78 | 17.73 | 26.25 | 23.81 | 34.29 |
| *Zero-shot Methods* | | | | | | | | | | | |
| Kosmos-2 (Peng et al., 2023) | 2B | 10.43 | 28.61 | 21.18 | 33.33 | 17.77 | 28.46 | 21.43 | 21.25 | 14.29 | 23.17 |
| InstructBLIP (Dai et al., 2023) | 13B | 38.39 | 30.52 | 26.27 | 76.67 | 70.66 | 35.77 | 30.00 | 22.50 | 19.05 | 35.94 |
| LLava-V1.5 (Liu et al., 2024b) | 13B | 36.97 | 27.46 | 20.22 | 52.22 | 23.55 | 27.64 | 22.86 | 45.00 | 4.76 | 27.05 |
| CogVLM (Wang et al., 2024c) | 17B | 52.61 | 37.42 | 26.91 | 55.56 | 54.13 | 29.27 | 29.29 | 32.50 | 23.81 | 37.19 |
| Gemini (Team et al., 2023) | - | 73.93 | 41.25 | 31.21 | 56.67 | 71.49 | 62.60 | 30.71 | 27.50 | 28.57 | 45.17 |
| GPT4V (Achiam et al., 2023) | - | 80.09 | 54.66 | 43.95 | 87.78 | 67.77 | 82.11 | 42.14 | 43.75 | 42.86 | 56.95 |
| *Finetuning Methods* | | | | | | | | | | | |
| LLaMA-Adapter (Zhang et al., 2023) | 7B | 62.56 | 72.29 | 30.21 | 76.92 | 59.67 | 72.36 | 30.71 | 38.75 | 38.10 | 54.89 |
| LLaVA-V1.5 (Liu et al., 2024b) | 13B | 68.72 | 72.41 | 40.86 | 83.52 | 64.61 | 69.11 | 35.71 | 45.00 | 38.10 | 59.50 |
| CogVLM (Wang et al., 2024c) | 17B | 65.88 | 77.52 | 29.09 | 81.32 | 65.43 | 75.61 | 35.71 | 46.25 | 47.62 | 58.25 |
| MC-CoT$_{base}$ (Tan et al., 2024a) | 223M | 53.55 | 63.98 | 43.56 | 61.54 | 69.55 | 29.27 | 42.86 | 33.75 | 28.57 | 53.51 |
| MC-CoT$_{large}$ (Tan et al., 2024a) | 738M | 42.65 | 67.43 | 50.56 | 58.24 | 60.49 | 56.10 | 57.86 | 62.50 | 14.29 | 57.69 |
| Multimodal-CoT$_{base}$ (Zhang et al., 2024b) | 223M | 41.71 | 46.49 | 39.90 | 59.34 | 60.91 | 27.64 | 48.57 | 35.00 | 28.57 | 44.85 |
| **MIND$_{base}$ (Our)** | 223M | 72.04 | 63.09 | 43.31 | 82.22 | 65.29 | 44.72 | 49.29 | 61.25 | 33.33 | 57.38 |
| Improvement | - | 30.33 ⇑ | 16.60 ⇑ | 3.41 ⇑ | 22.88 ⇑ | 4.38 ⇑ | ⇑ 17.08 | 0.72 ⇑ | 26.25 ⇑ | 4.76 ⇑ | 12.53 ⇑ |
| Multimodal-CoT$_{large}$ (Zhang et al., 2024b) | 738M | 45.50 | 50.19 | 43.56 | 63.74 | 64.61 | 33.33 | 40.71 | 61.25 | 28.57 | 48.73 |
| **MIND$_{large}$ (Our)** | 738M | 79.62 | 66.41 | 48.57 | 81.11 | 69.42 | 51.22 | 50.71 | 61.25 | 47.62 | 61.56 |
| Improvement | - | 34.12 ⇑ | 16.22 ⇑ | 5.01 ⇑ | 17.37 ⇑ | 4.81 ⇑ | 17.89 ⇑ | 10.00 ⇑ | 0.00 ⇑ | 19.05 ⇑ | 12.83 ⇑ |

verify the complementarity between P2CL and MCA, whose collaboration jointly optimizes semantic coherence.

*2) Investigating the Impact of Rationale Quality and Quantity.* On the one hand, to evaluate the impact of the rationale quality, we conduct a comparative analysis across the DeepSeek series (Liu et al., 2024a; Guo et al., 2025), Qwen series (Yang et al., 2025a). From Table 5, with the improvement of the generation model's capability, the performance consistently increases. Specifically, when using rationales generated by DeepSeek-R1-Qwen8B, the accuracy reaches 91.49%. Further employing the more powerful Qwen3-235B-22A for rationale generation boosts the accuracy to 91.61%, indicating that rationales produced by stronger models possess higher semantic completeness and logical consistency. In comparison, rationales generated by DeepSeek V3 yield limited improvement. Notably, Deepseek-R1 has only one-third the Rationales of the others. Ultimately, the *"Final Mix"* achieves the best performance, which verifies that the multi-source generation strategy can effectively leverage the complementary strengths of different models in semantic coverage and logical reasoning. On the other hand, we further evaluate MIND under different rationale scales. From Table 6, the model's accuracy steadily improves as the number of rationales increases, though the gain gradually saturates. Specifically, compared to *"Original (21K)"*, using more rationales results in a performance improvement of 1.13% - 2.00%. In addition, when the number of rationales reaches ×1000 (21M), the performance increase tends to level off, indicating that the model's performance is close to saturation under ultra-large-scale

rationales. Notably, the training cost does not scale proportionally with the number of rationales, as rationales are randomly sampled for each question during training rather than being fully enumerated.

*3) Investigation of Epochs, Caption Generation, and Visual Feature Extraction.* From Figure 4, model performance improves steadily with an increasing number of training epochs. The performance gain is pronounced when the number of epochs increases from 20 to 50, while the improvement gradually plateaus beyond 50 to 200 epochs. In this work, we adopt 200 epochs as the default setting for all experiments. Regarding image caption generation, we compare InstructBLIP (Dai et al., 2023) and Qwen2.5-VL (Bai et al., 2025). As shown in Figure 4, larger models with stronger semantic and visual alignment abilities (e.g., Qwen2.5-VL-72B) produce more complete and coherent captions, thereby improving rationale consistency and answer reliability. For visual feature extraction, the results are shown in Table 7. On the one hand, larger-scale visual encoders provide more robust visual grounding for reasoning. For instance, SAM-Huge (Kirillov et al., 2023) and DINOv2-Giant (Oquab et al., 2023) preserve detailed structures while reinforcing global semantic coherence, thereby improving reasoning accuracy. On the other hand, multimodal models with language enhancement mechanisms (such as CLIP-L14-336 (Radford et al., 2021), BLIP-Large (Li et al., 2022) and BLIP2-flan-t5-xxl (Li et al., 2023)) establish a closer semantic mapping between vision and text, enabling the model to more accurately align the semantics of the image and text when generating rationales

*Table 4.* Break-Down ablation on the ScienceQA dataset.

| Schemes | P2CL | MCA | *Acc.* | *Improve.* |
|---|---|---|---|---|
| Baseline | ✗ | ✗ | 90.29 | - |
| MIND$_{base}$ w/o P2CL | ✗ | ✓ | 90.36 | 0.07 ⇑ |
| MIND$_{base}$ w/o MCA | ✓ | ✗ | 92.15 | 1.86 ⇑ |
| **MIND$_{base}$ (Ours)** | ✓ | ✓ | **92.29** | **2.00 ⇑** |

*Table 5.* Performance analysis of generating rationales using different large models on the ScienceQA dataset.

| Large Models | *Acc.* | Large Models | *Acc.* |
|---|---|---|---|
| Baseline | 90.29 | DeepseekR1-Qwen8B | 91.49 |
| Qwen3-235B-22A | 91.61 | Deepseek-R1 | 90.73 |
| DeepseekV3 | 90.87 | **Final Mix** | **92.29** |

*Table 6.* Performance analysis of different numbers of rationales on the ScienceQA dataset. "$\times N$" denotes $N$-fold expansion.

| Rationales | *Acc.* | Rationales | *Acc.* |
|---|---|---|---|
| Original (21K) | 90.29 | ×100 (2.1M) | 91.58 (1.29 ⇑) |
| ×10 (210K) | 91.42 (1.13 ⇑) | ×500 (10.5M) | 91.91 (1.62 ⇑) |
| ×50 (1.05M) | 91.56 (1.27 ⇑) | **×1000 (21M)** | **92.29 (2.00 ⇑)** |

*Table 7.* Performance analysis of visual feature extraction methods on the ScienceQA dataset. **VFE**: Visual feature extraction

| VFE Models | *Acc.* | VFE Models | *Acc.* |
|---|---|---|---|
| SAM-Base | 90.80 | SAM-Huge | 91.25 |
| DINOv2-Large | 91.06 | DINOv2-Giant | 91.51 |
| CLIP-B16 | 91.51 | CLIP-L14-336 | 91.98 |
| BLIP-Large | 91.49 | **BLIP2-flan-t5-xxL** | **92.29** |

*Table 8.* Performance comparison of MIND with different settings on the ScienceQA dataset. The left and right arrows indicate the injection operations at the input and supervision of P2CL-II, respectively. **Pos:** Positive rationales. **Neg:** Negative Rationales.

| Schemes | *Acc.* | Schemes | *Acc.* |
|---|---|---|---|
| MIND w/o P2CL-I | 91.63 | MIND w/o P2CL-II | 90.36 |
| Pos → N/A | 91.20 | Pos → Pos | 91.20 |
| Neg → N/A | 90.64 | Neg → Pos | 91.72 |
| Pos→ N/A or Neg → N/A | 91.37 | Pos → Pos or Neg → N/A | 91.39 |
| Pos→ N/A or Neg → Pos | 91.96 | **Pos → Pos or Neg → Pos** | **92.29** |

and answers. This demonstrates that higher-level semantic representation and stronger cross-modal alignment capabilities are key factors in improving reasoning performance.

### 4.5. Discussion

*1) "Understand → Rethink → Correct".* From Table 8, first, removing the P2CL-I alone leads to a 0.66% (from 92.29% to 91.63%) performance drop, indicating that semantic consistency learning lays the foundation for model understanding. At the same time, removing the P2CL-II stage alone results in a significant performance decrease of 1.93% (from 92.29% to 90.36%), which validates the core role of active logic discrimination and correction. Second, when the model outputs only an answer, providing a positive rationale input is more effective than a negative one, because positive semantics help stabilize logical structure,

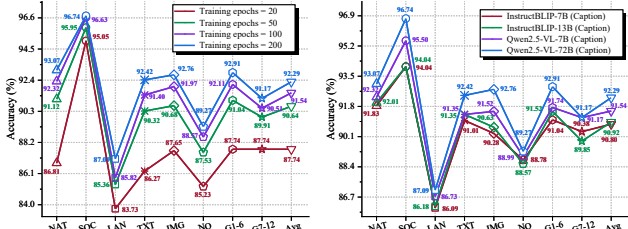

*Figure 4.* Performance analysis of Epochs and Caption generation methods on the ScienceQA dataset.

whereas negative rationales can drift without correction signals. Interestingly, the latter still outperforms removing P2CL-II entirely. This indicates that negative rationales provide adversarial cognitive stimulation, prompting the model to develop the ability to distinguish between correct and incorrect logic. Third, injecting a negative rationale input while supervising with a positive one yields notable gains. Specifically, compared to *"Neg → N/A"*, *"Neg → Pos"* directly improves 1.08% (from 90.64% to 91.72%); compared to *"Pos → N/A or Neg → N/A"*, *"Pos → N/A or Neg → Pos"* directly improves 0.59% (from 91.37 to 91.96). This demonstrates that the joint design of negative rationale input and positive rationale supervision enables explicit logic correction, allowing the model to transition from error identification to logic reconstruction. Finally, jointly using positive/negative rationale inputs with positive supervision yields the best results. In summary, the P2CL strategy effectively realizes a progressive reasoning mechanism of "Understand → Rethink → Correct" through semantic understanding of P2CL-I and logic discrimination and correction of P2CL-II, enabling the model to develop stronger logical stability and error correction capabilities, and achieving a paradigm evolution from passive imitation-based reasoning to active discriminative reasoning.

*2) Excellent Robustness to Unseen Interference Types.* We further randomly sample 100 samples from each dataset and use Qwen3-235B-22A to construct five additional unseen error types in addition to the original error type: *Plausible but Wrong*, *Irrelevant Distractor*, *Incomplete Reasoning*, *Selective Evidence*, and *Logical Misbinding*. The results are shown in Table 9, where *"Normal"* denotes normal reasoning without interference. Compared to *"Normal"*, Multimodal-CoT$_{base}$ shows a significant performance decline under all six perturbations, including the original error type used in RAD to generate negative samples, while MIND$_{base}$ remains stable and even shows a slight improvement on M$^3$CoT. This indicates that MIND is more robust to both seen and unseen reasoning biases and directly supports the effectiveness of the active logic discrimination and correction mechanism in P2CL-II. This further highlights its potential for real-world applications.

*3) Excellent Compatibility and Generalizability.* MIND is a paradigm-level enhancement, rather than being tied to a spe-

*Table 9.* Evaluation on input perturbation task across multiple datasets (100 test samples per dataset).

| Dataset | Methods | Normal | Original Error Type | Plausible but Wrong | Irrelevant Distractor | Incomplete Reasoning | Selective Evidence | Logical Misbinding |
|---|---|---|---|---|---|---|---|---|
| ScienceQA | Multimodal-CoT$_{base}$ | 84.0% | 34.0% | 43.0% | 47.0% | 66.0% | 35.0% | 41.0% |
| | **MIND$_{base}$** | **91.0%** | **90.0%** | **91.0%** | **88.0%** | **93.0%** | **89.0%** | **91.0%** |
| A-OKVQA | Multimodal-CoT$_{base}$ | 52.0% | 16.0% | 19.0% | 16.0% | 37.0% | 4.0% | 19.0% |
| | **MIND$_{base}$** | **68.0%** | **67.0%** | **68.0%** | **67.0%** | **67.0%** | **65.0%** | **69.0%** |
| M$^3$CoT | Multimodal-CoT$_{base}$ | 42.0% | 25.0% | 34.0% | 14.0% | 46.0% | 8.0% | 13.0% |
| | **MIND$_{base}$** | **55.0%** | **59.0%** | **55.0%** | **59.0%** | **60.0%** | **56.0%** | **60.0%** |

*Table 10.* Compatibility and generalization assessment of MIND. The left and right arrows denote training and testing.

| Setting | M$^3$CoT (Train set)–>M$^3$CoT (Test set) | | | M$^3$CoT (Train set)–>ScienceQA (Test set) | | |
|---|---|---|---|---|---|---|
| Models | Qwen2.5-VL-7B | Qwen3-VL-8B | Qwen3.5-9B | Qwen2.5-VL-7B | Qwen3-VL-8B | Qwen3.5-9B |
| Origin (No SFT) | 60.96 (+24.89) | 65.83 (+22.31) | 70.02 (+18.63) | 82.74 (+6.32) | 90.57 (+1.18) | 90.71 (+1.39) |
| Baseline (MIND w/o MCA, P2CL) | 77.31 (+8.54) | 77.74 (+10.40) | 75.97 (+12.68) | 83.80 (+5.26) | 88.16 (+3.59) | 85.00 (+7.10) |
| **MIND (Final)** | **85.85** | **88.14** | **88.65** | **89.06** | **91.75** | **92.10** |

*Table 11.* Rationale quality assessment on multiple datasets. Total score: 10. The scores are "ScienceQA / A-OKVQA / M$^3$CoT".

| Metric | Overall | Correctness | Relevance | Coherence | Solving Process |
|---|---|---|---|---|---|
| Multimodal-CoT$_{base}$ | 8.12 / 6.85 / 3.63 | 7.49 / 5.49 / 2.22 | 8.64 / 8.03 / 4.69 | 8.35 / 7.88 / 4.12 | 8.00 / 6.00 / 3.47 |
| **MIND$_{base}$ (P2CL-I)** | 8.40 / 7.47 / 5.34 | 8.01 / 6.48 / **3.94** | 8.73 / 8.30 / 6.40 | 8.55 / 7.99 / 5.85 | 8.32 / 7.09 / 5.18 |
| **MIND$_{base}$ (P2CL-II)** | **8.53 / 7.94 / 5.54** | **8.08 / 6.71** / 3.84 | **8.88 / 8.98 / 6.81** | **8.71 / 8.60 / 6.17** | **8.44 / 7.46 / 5.34** |

cific model. P2CL explicitly models the "Understand → Rethink → Correct" process, while MCA further strengthens the semantic discriminative boundaries in the multi-rationale semantic space. MIND focuses on the training paradigm and has excellent transferability. From Table 10, we have extended MIND to Qwen2.5-VL, Qwen3-VL, and Qwen3.5 and evaluated it on the difficult M$^3$CoT dataset. Compared to the original model without SFT, MIND improves performance by 18.64% - 24.89%. Compared to the baseline with MCA and P2CL removed, it still improves performance by 8.54% - 12.68%. This further demonstrates that MIND can naturally migrate to larger and stronger base models and continue to deliver significant benefits. In addition, to verify the generalization of MIND on out-of-distribution tasks, we use the M$^3$CoT dataset for training and the ScienceQA dataset for testing. From Table 10, compared to the original model without SFT, the *"Baseline"* often impairs cross-dataset generalization, while our MIND still consistently achieves improvements of 1.18% - 6.32%. Furthermore, compared to the corresponding *"Baseline"*, MIND achieves a significant improvement of 3.59% - 7.10%. These results indicate that MIND retains stronger generalization ability and reasoning capability in cross-dataset scenarios.

*4) Quantitative Assessment of Rationale Quality.* To further verify the rationale quality generated by MIND, in addition to the qualitative analysis in Figures 6 to 8, we conduct an additional quantitative evaluation. To avoid subjective human factors, we use a larger judge model (Qwen3-VL-30B-A3B) to score across four dimensions (*Correctness*, *Relevance*, *Coherence*, *Solving Process*). Each dimension is further divided into five score levels with explicit prompt definitions.

Considering that MIND explicitly models discrimination and correction through P2CL-II, we report both *"MIND$_{base}$ (P2CL-I)"* and *"MIND$_{base}$ (P2CL-II)"*. As shown in Table 11, compared with *Multimodal-CoT$_{base}$*, *MIND$_{base}$ (P2CL-I)* achieves higher scores across all four dimensions, with an overall improvement of 0.28 - 1.71. Furthermore, *MIND$_{base}$ (P2CL-II)* continues to improve across almost all dimensions. This shows that MIND improves not only final answer accuracy, but also rationale quality and the model's response to flawed reasoning cues.

## 5. Conclusion

This paper introduces the MIND reasoning framework for MLLMs, aiming to endow the model with a human-like cognitive pattern of "Understand → Rethink → Correct", thereby achieving a paradigm evolution from passive imitation-based reasoning to active discriminative reasoning. Specifically, we propose a RAD paradigm, which provides a unified, scalable and high-quality data foundation. At the same time, a P2CL strategy is designed that enables multi-rationale positive learning and active logic discrimination and correction. In addition, a MCA optimization strategy is proposed to further enhance the model's logical consistency and discriminative capability. Experimental results show that our MIND achieves SOTA performance on the ScienceQA, A-OKVQA, and M$^3$CoT datasets, exhibiting stronger generalization and interpretability. We hope that this work can draw attention to the research on multi-rationale discrimination reasoning and promote MLLMs toward a new stage of intelligent reasoning.

## Acknowledgements

This work is partially supported by the ITSP Platform Project (No. ITS/600/24FP), the Centre for Perceptual and Interactive Intelligence, a CUHK-led InnoCentre under the InnoHK initiative of the Innovation and Technology Commission of the Hong Kong Special Administrative Region Government, the Hong Kong RGC Strategic Topics Grant (No. STG1/E-403/24-N), the CUHK-CUHK(SZ)-GDST Joint Collaboration Fund (No. YSP26-4760949), the University of Sydney-Chinese University of Hong Kong Ignition Grant (No. G232965), and the Institute Basic Research Program (No. E429040701).

## Impact Statement

This paper presents work whose goal is to advance the field of Machine Learning. There are many potential societal consequences of our work, none of which we feel must be specifically highlighted here.

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

## A. Positive and Negative Prompts.

In this section, we present positive and negative prompts in detail. Details are as follows:

*Positive Prompts:* "{*Task-related information*}"\n You are an intelligent agent with both perception and reasoning abilities. Based on the given context, please make random content adjustments to "{*Solution*}" within a range of 10% to 50% while ensuring that the semantics remain unchanged. Please output {*Repeat_number*} different solutions. Each output format is "Adjusted Solution:". Use "\n\n~~~\n\n" to separate them. The output must be in the required format.

*Negative Prompts:* "{*Task-related information*}"\n You are an intelligent agent with both perception and reasoning abilities. "{*Solution*}" is the explanation for the above problem. Based on the given context, please make minor edits to "{*Solution*}" to reverse its meaning and ensure the correct answer cannot be logically derived, while keeping most of the original words and structure intact. Please output {*Repeat_number*} different solutions. Each output format is "Negative Solution:". Use "\n\n~~~\n\n" to separate them. The output must be in the required format.

Specifically, {*Task-related information*} refers to individual sample information such as questions and options. {*Solution*} refers to the Rationale. {*Repeat_number*} refers to the number of rationales that can be generated in a single API call, facilitating batch generation. "\n\n~~~\n\n" is the predefined delimiter.

## B. More Quantitative Experiments.

In this section, we have added more quantitative experiments, such as more break-down ablation experiments, the hyperparameters in the MCA optimization strategy and so on. Details are as follows:

*1) More Break-Down Ablation.* To more comprehensively analyze the independent contributions and synergistic gains of each core component, we conduct systematic ablation studies on the P2CL strategy and MCA optimization strategies across ScienceQA, A-OKVQA, and M³CoT datasets. The experimental results are shown in Table 12. First, compared to the *"Baseline"*, using only the MCA optimization strategy results in an accuracy improvement of 0.07% (from 90.29% to 90.36%) - 3.02% (from 52.67% to 55.69%), indicating that the MCA optimization strategy can enhance reasoning consistency and improving the model's discrimination sensitivity. Second, compared to the *"Baseline"*, using only the P2CL strategy alone results in an accuracy improvement of 1.86% (from 90.29% to 92.15%) - 4.28% (from 65.85% to 70.13%), indicating that multi-rationale positive learning and active logic discrimination and correction can effectively strengthen reasoning robustness. Finally, the MIND reasoning framework, combining the MCA and P2CL strategies, achieves the best results on each dataset. Specifically, compared to the "Baseline", jointly applying MCA and P2CL strategies improves accuracy by 2.00% (from 90.29% to 92.29%) - 4.72% (from 65.85% to 70.57%). This illustrates the complementarity between the P2CL and MCA strategies. The P2CL strategy constructs reliable logic and enables error-correctable reasoning, whereas MCA optimization strategy sharpens semantic boundaries and strengthens contrastive discrimination. Their collaboration jointly optimizes semantic coherence and logical discrimination, thereby significantly enhancing the model's reasoning capability.

*Table 12.* Break-Down ablation experiments on ScienceQA, A-OKVQA, and M³CoT datasets.

| Schemes | P2CL | MCA | ScienceQA dataset | | A-OKVQA dataset | | M³CoT dataset | |
|---|---|---|---|---|---|---|---|---|
| | | | *Accuracy* | *Improve.* | *Accuracy* | *Improve.* | *Accuracy* | *Improve.* |
| Baseline | ✗ | ✗ | 90.29 | - | 65.85 | - | 52.67 | - |
| MIND$_{base}$ w/o P2CL | ✗ | ✓ | 90.36 | 0.07 ⇑ | 68.03 | 2.18 ⇑ | 55.69 | 3.02 ⇑ |
| MIND$_{base}$ w/o MCA | ✓ | ✗ | 92.15 | 1.86 ⇑ | 70.13 | 4.28 ⇑ | 56.34 | 3.67 ⇑ |
| **MIND$_{base}$ (Ours)** | ✓ | ✓ | **92.29** | **2.00 ⇑** | **70.57** | **4.72 ⇑** | **57.38** | **4.71 ⇑** |

*2) Exploration of Hyperparameters in the MCA Optimization Strategy.* We conduct a systematic analysis under various parameter settings. From Figure 5, the experimental results with different values of $m$ and $\alpha$ demonstrate that the MCA optimization strategy has remarkable stability. In particular, when $m \in [0.1 - 0.4]$ and $\alpha \in [0.5 - 5.0]$, the model

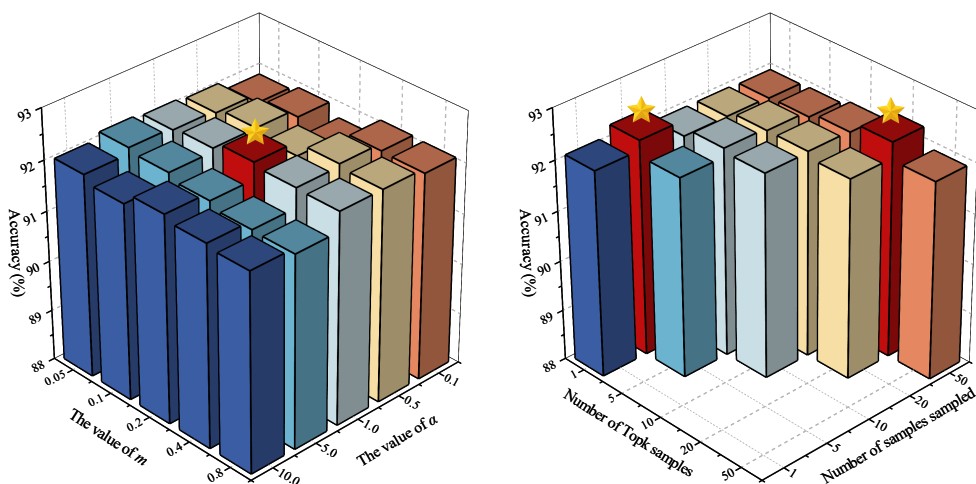

*Figure 5.* Exploring the hyperparameters of the MCA optimization strategy on the ScienceQA dataset. **Left:** $m$ and $\alpha$. **Right:** Sampled rationales and Top-k. Red bars denote the optimal.

performance remains stable and varies smoothly. The optimal performance is achieved when $m = 0.2$ and $\alpha = 1$. At the same time, from the system experiments on the sampling number and Top-k setting in Figure 5, compared to the scheme that uses all samples in the outermost row, selecting the Top-k hard rationales under the same sampling scale can achieve higher performance. Specifically, the model achieves its best results when the *(sampled rationales, Top-k)* settings are *(5, 1)* and *(50, 20)*, respectively. These findings indicate that maintaining rationale diversity while moderately focusing on hard rationales can effectively enhance the discriminative capability and semantic boundary modeling. Notably, MCA uses the same hyperparameter settings in all other experiments.

*3) Comparison with InfoNCE Loss.* From Table 13, it shows that InfoNCE with different temperature coefficients consistently underperforms the margin-based contrastive loss. Together with the hyperparameter analysis in Figure 5, these results demonstrate that the current design achieves better objective alignment, optimization effectiveness, and robustness. From the perspective of optimization objectives, MCA explicitly encourages hard positive aggregation and hard negative separation, directly shaping the local discriminative boundary in the multi-rationale semantic space. In contrast, InfoNCE mainly relies on global normalization-based competition, making it more sensitive to temperature settings and sample quality. Therefore, the margin-based contrastive loss is better aligned with the goal of MCA.

*Table 13.* Performance comparison of different loss settings across multiple datasets. $T$ denotes the temperature.

| Loss setting | ScienceQA dataset | A-OKVQA dataset | M$^3$CoT dataset |
|---|---|---|---|
| InfoNCE (T=0.05) | 91.89 (+0.40) | 69.52 (+1.05) | 56.08 (+1.30) |
| InfoNCE (T=0.07) | 92.03 (+0.26) | 70.13 (+0.44) | 56.56 (+0.82) |
| InfoNCE (T=0.1) | 92.05 (+0.24) | 69.87 (+0.70) | 56.34 (+1.04) |
| **Margin-based Contrastive** | **92.29** | **70.57** | **57.38** |

*Table 14.* Performance comparison of Multimodal CoT and MIND with the same visual encoder and caption generator.

| Schemes | BLIP2-flan-t5-xxl + InstructBLIP-7B caption | Vit-large-patch32-384 + Qwen2.5-VL-72B caption | BLIP2-flan-t5-xxl + Qwen2.5-VL-72B caption |
|---|---|---|---|
| Multimodal-CoT$_{base}$ | 86.98 (+3.82) | 87.53 (+4.24) | 87.79 (+4.50) |
| **MIND$_{base}$** | **90.80** | **91.77** | **92.29** |

*4) Comparison of Multimodal-CoT and MIND with the Same Visual Encoder and Caption Generator.* In the MIND model, we use frozen BLIP2-flan-t5-xxl and Qwen2.5-VL-72B as the visual encoder and caption generator, respectively. To more fairly compare the effectiveness of the Multimodal-CoT and MIND frameworks, we further conduct a more rigorous matched-setting comparison. As shown in Table 14, while keeping the visual encoder and caption generator consistent, MIND$_{base}$ still achieves a stable performance improvement of 3.82% - 4.50% compared to Multimodal-CoT$_{base}$. The significant performance improvement indicates that the gains cannot be simply attributed to the assistance of stronger

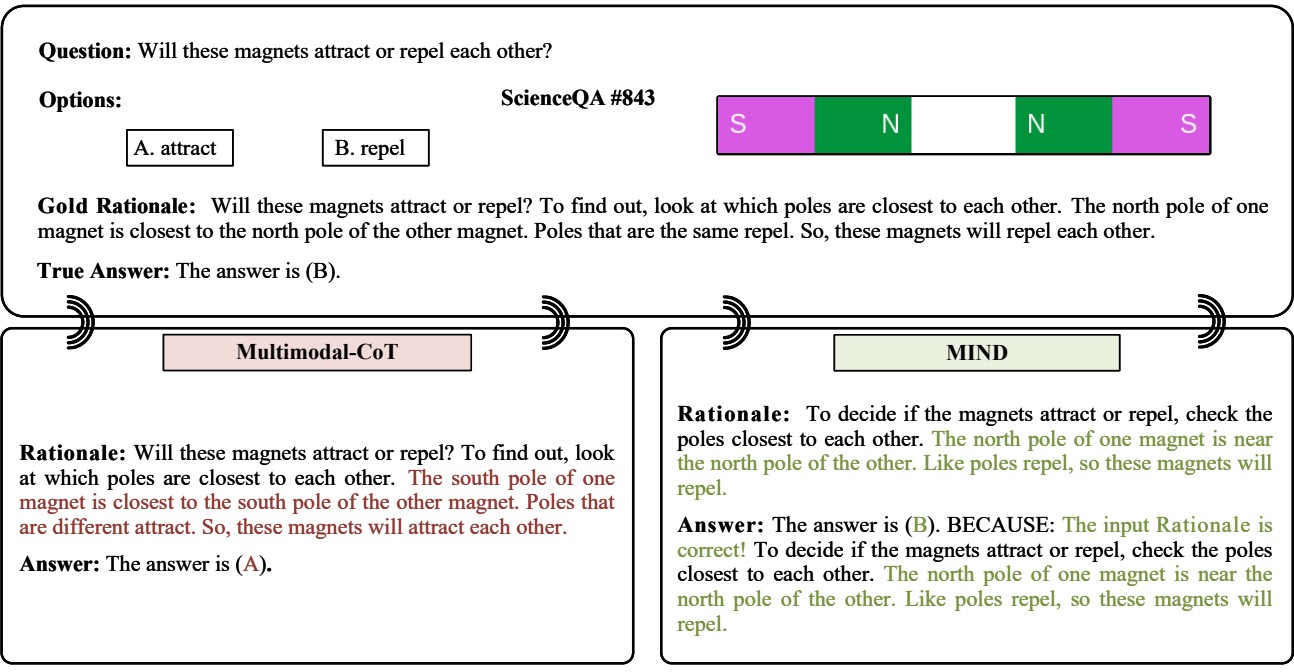

*Figure 6.* Reasoning case comparison between MIND$_{base}$ and Multimodal-CoT$_{base}$ on selected samples from the ScienceQA dataset. Red denotes incorrect reasoning, green denotes correct reasoning, and "#" denotes the Question ID.

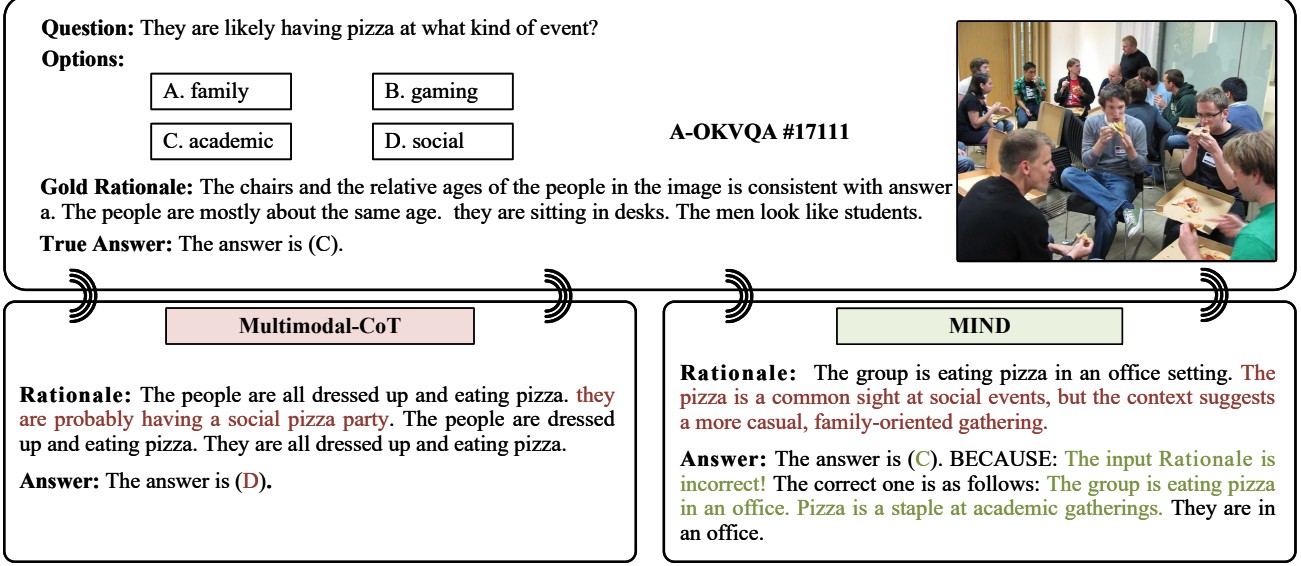

*Figure 7.* Reasoning case comparison between MIND$_{base}$ and Multimodal-CoT$_{base}$ on selected samples from the A-OKVQA dataset. Red denotes incorrect reasoning, green denotes correct reasoning, and "#" denotes the Question ID.

external modules, but also come from the proposed training framework itself.

## C. Qualitative Analysis

In this section, we have provided a detailed qualitative analysis. Specifically, on the one hand, we qualitatively compare Multimodal-CoT (Zhang et al., 2024b) and MIND. On the other hand, we conduct a qualitative analysis of MIND's Error-Correction capability. Details are as follows:

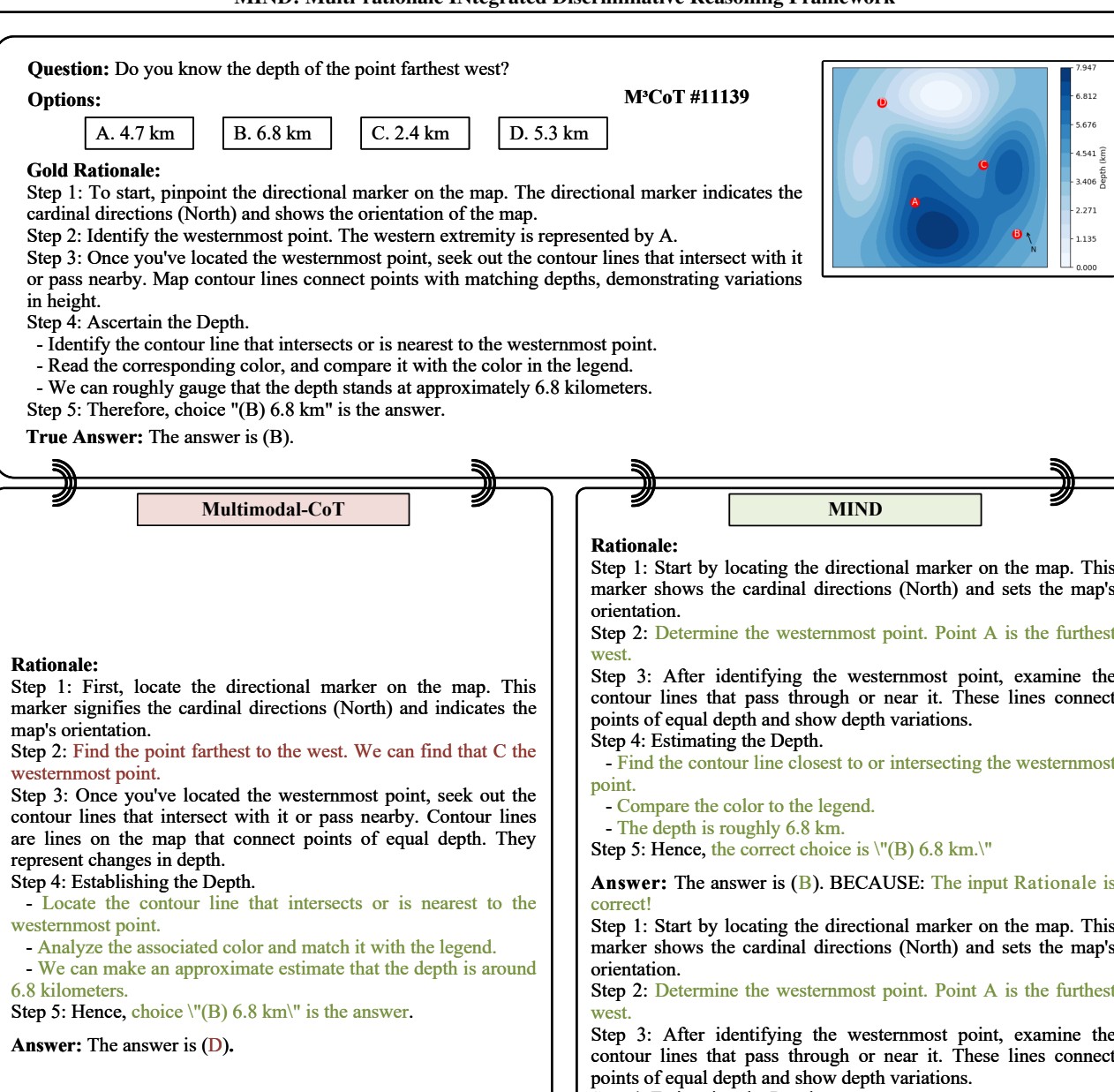

*Figure 8.* Reasoning case comparison between MIND$_{base}$ and Multimodal-CoT$_{base}$ on selected samples from the M$^3$CoT dataset. Red denotes incorrect reasoning, green denotes correct reasoning, and "#" denotes the Question ID.

*1) Comparison Between Multimodal-CoT and MIND.* To more intuitively demonstrate the superior interpretability and logical consistency of MIND in multimodal reasoning, we select one sample from each of the ScienceQA, A-OKVQA, and M$^3$CoT datasets for qualitative comparison with Multimodal-CoT. The visualization results are shown in Figures 6 to 8. We observe that prior Multimodal-CoT often exhibit issues such as logical confusion and conclusion drift in scenes with visual ambiguity or semantic interference. In contrast, MIND learns a multi-rationale semantic distribution during training, enabling it to adaptively focus on key semantic cues under complex inputs and produce reasoning results with stronger causal coherence. At the same time, our MIND uses active logic discrimination and correction in the Phase II (P2CL-II) of learning. When the input contains logical errors or misleading rationales, Multimodal-CoT tends to follow the flawed

**Question:** Complete the text to describe the diagram. Solute particles moved in both directions across the permeable membrane. But more solute particles moved across the membrane (). When there was an equal concentration on both sides, the particles reached equilibrium.

**Options:**

A. to the left than to the right    B. to the right than to the left

**ScienceQA #13736**

**Gold Rationale:** Look at the diagram again. It shows you how the solution changed during the process of diffusion. Before the solute particles reached equilibrium, there were 4 solute particles on the left side of the membrane and 2 solute particles on the right side of the membrane. When the solute particles reached equilibrium, there were 3 solute particles on each side of the membrane. There was 1 more solute particle on the right side of the membrane than before. So, for the solute particles to reach equilibrium, more solute particles must have moved across the membrane to the right than to the left.

**True Answer:** The answer is (B).

**Phase I: Rationale generation**

**Rationale:** The diagram demonstrates diffusion. Initially, the left side had 4 particles, and the right side had 2. At equilibrium, both sides had 4 particles. The right side's gain of 1 particle indicates more movement to the right than to the left.

**Phase II: Answer generation**

**Answer:** The answer is (B). BECAUSE: The input Rationale is incorrect! The correct one is as follows: The diagram illustrates diffusion. Initially, 4 particles were on the left and 2 on the right. At equilibrium, both sides had 3 particles, indicating more particles moved to the right.

---

**Question:** Which of the following organisms is the decomposer in this food web?

**Options:**

A. green algae    B. water mold    C. golden algae

**ScienceQA #1392**

**Gold Rationale:** Decomposers help break down dead organisms into simpler matter, such as nutrients. These nutrients can then help plants and other organisms grow. In a food web, there is an arrow pointing from another organism to a decomposer. There are no arrows pointing from a decomposer to another organism. The water mold does not have arrows pointing from it to other organisms. So, the water mold is a decomposer. The bacteria do not have arrows pointing from them to other organisms. So, the bacteria are decomposers. The green algae has arrows pointing from it. So, the green algae is not a decomposer. The golden algae has an arrow pointing from it. So, the golden algae is not a decomposer.

**True Answer:** The answer is (B).

**Phase I: Rationale generation**

**Rationale:** Decomposers break down dead organisms into simpler nutrients, supporting plants and other organisms. In food webs, arrows point to decomposers, not from them. Green algae has arrows pointing to it, so it's not a decomposer. Water mold has arrows pointing to it, so it's not a decomposer. Golden algae has arrows pointing to it, so it's not a decomposer.

**Phase II: Answer generation**

**Answer:** The answer is (B). BECAUSE: The input Rationale is incorrect! The correct one is as follows: Decomposers break down dead organisms into simpler nutrients, supporting plants and other organisms. In food webs, arrows point to decomposers, but not from them. Green algae has outgoing arrows, so it's not a decomposer. Water mold has no outgoing arrows, so it's a decomposer. Golden algae has outgoing arrows, so it's not a decomposer. Water mold has no outgoing arrows, so it's a decomposer.

*Figure 9.* Some error correction cases of MIND$_{base}$ on the ScienceQA dataset. Red denotes incorrect reasoning, green denotes correct reasoning, and "#" denotes the Question ID. The proposed MIND reasoning framework demonstrates excellent error correction capabilities.

semantic direction and produce unreasonable conclusions. In comparison, MIND can actively detect reasoning deviations and reconstruct correct logic aligned with factual and visual semantics.

*2) MIND's Error-Correction Capability.* To more intuitively and comprehensively demonstrate how MIND can reflect on and correct flawed logic in the rationales generated during Phase I (P2CL-I), we conduct qualitative analyses on some samples from each of the ScienceQA, A-OKVQA, and M³CoT datasets. The visualization results are shown in Figures 9 to 11. In the cases presented, the *Rationale* generated in Phase I contains varying degrees of misleading logic, such as incorrect factual associations and logical leaps. For these *Rationales* with reasoning biases, our MIND exhibits strong

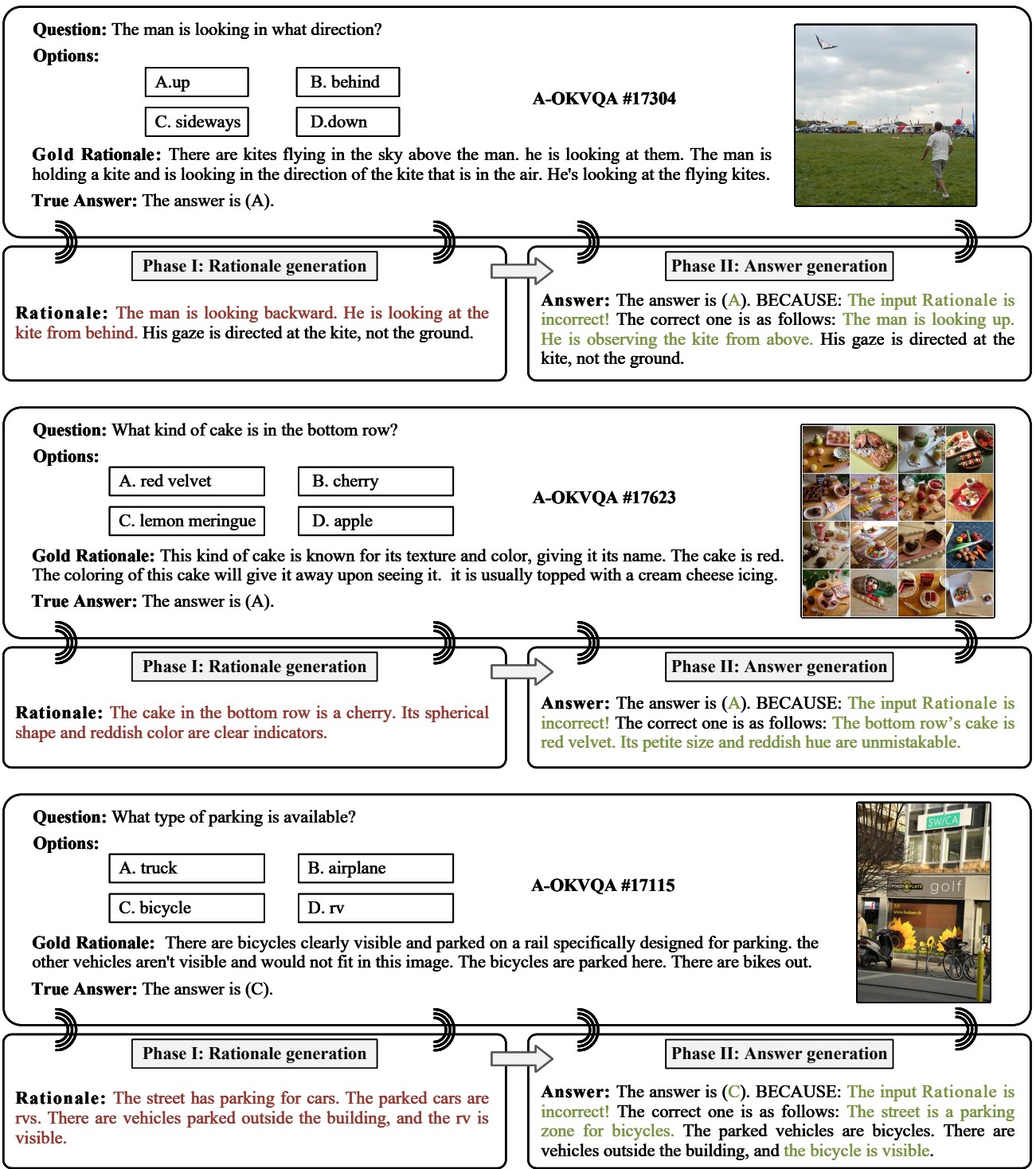

**Question:** The man is looking in what direction?

**Options:**

A.up          B. behind

C. sideways          D.down

**A-OKVQA #17304**

**Gold Rationale:** There are kites flying in the sky above the man. he is looking at them. The man is holding a kite and is looking in the direction of the kite that is in the air. He's looking at the flying kites.

**True Answer:** The answer is (A).

**Phase I: Rationale generation**

**Rationale:** The man is looking backward. He is looking at the kite from behind. His gaze is directed at the kite, not the ground.

**Phase II: Answer generation**

**Answer:** The answer is (A). BECAUSE: The input Rationale is incorrect! The correct one is as follows: The man is looking up. He is observing the kite from above. His gaze is directed at the kite, not the ground.

**Question:** What kind of cake is in the bottom row?

**Options:**

A. red velvet          B. cherry

C. lemon meringue          D. apple

**A-OKVQA #17623**

**Gold Rationale:** This kind of cake is known for its texture and color, giving it its name. The cake is red. The coloring of this cake will give it away upon seeing it. it is usually topped with a cream cheese icing.

**True Answer:** The answer is (A).

**Phase I: Rationale generation**

**Rationale:** The cake in the bottom row is a cherry. Its spherical shape and reddish color are clear indicators.

**Phase II: Answer generation**

**Answer:** The answer is (A). BECAUSE: The input Rationale is incorrect! The correct one is as follows: The bottom row's cake is red velvet. Its petite size and reddish hue are unmistakable.

**Question:** What type of parking is available?

**Options:**

A. truck          B. airplane

C. bicycle          D. rv

**A-OKVQA #17115**

**Gold Rationale:** There are bicycles clearly visible and parked on a rail specifically designed for parking. the other vehicles aren't visible and would not fit in this image. The bicycles are parked here. There are bikes out.

**True Answer:** The answer is (C).

**Phase I: Rationale generation**

**Rationale:** The street has parking for cars. The parked cars are rvs. There are vehicles parked outside the building, and the rv is visible.

**Phase II: Answer generation**

**Answer:** The answer is (C). BECAUSE: The input Rationale is incorrect! The correct one is as follows: The street is a parking zone for bicycles. The parked vehicles are bicycles. There are vehicles outside the building, and the bicycle is visible.

*Figure 10.* Some error correction cases of MIND$_{base}$ on the A-OKVQA dataset. Red denotes incorrect reasoning, green denotes correct reasoning, and "#" denotes the Question ID. The proposed MIND framework demonstrates excellent error correction capabilities.

reflective and corrective abilities in Phase II (P2CL-II). The model can detect key semantic conflicts that are inconsistent with the image, question, or ground-truth answer, and then generate revised Rationales that are both visually grounded and logically coherent. From Figures 9 to 11, in all scenarios including scientific reasoning (ScienceQA), open-domain common sense reasoning (A-OKVQA), and cross-domain complex reasoning (M³CoT), our MIND is able to recover the correct logical trajectory, further validating its human-like reasoning pattern of "Understand → Rethink → Correct". MIND not

**Question:** What does the arrangement of the geometric shapes in the image represent?

**Options:**

A. A car    B. A boat

C. A bird    D. A house

M³CoT #10742

**Gold Rationale:** The image consists of five different colored geometrical shapes arranged to represent the shape of a bird. The lemon-colored shape represents the neck, while the amber shape represents the body, and the orange shape represents the head. The mint shape forms the wing while the black shape is the tail feather. Therefore, the correct answer is C) A bird.

**True Answer:** The answer is (C).

**Phase I: Rationale generation**

**Rationale:** The image features a Tangram puzzle arranged to resemble a bird. The body is represented by the ivory piece, the wings by the black piece, the legs by the pewter piece, and the tail by the ivory piece. Consequently, the correct answer is option b) A boat.

**Phase II: Answer generation**

**Answer:** The answer is (C). BECAUSE: The input Rationale is incorrect! The correct one is as follows: The image displays a Tangram puzzle arranged to resemble a bird. The ivory piece is the body, the black piece the wings, the pewter piece the legs, and the ivory piece the tail. Consequently, the correct answer is option c) A bird.

**Question:** What kind of place is this most likely to be?

**Options:**

A. A motorcycle gang meetup spot

B. A shopping mall

C. A park

D. A restaurant

M³CoT #9153

**Gold Rationale:** The motorcycles lined up on the street together suggest that this place is a gathering spot for a motorcycle gang. There are many people here. The booths lined up in the background suggest that there are food stalls. Many green trees in the background suggest that this place icould be a park. Based on the inferences, it is most likely that this is a park (C) rather than a motorcycle gang meetup spot, a shopping mall, or a restaurant.

**True Answer:** The answer is (C).

**Phase I: Rationale generation**

**Rationale:** The motorcycles parked in a row on the pavement, indicating a motorcycle-centric setting, and the green stalls positioned nearby suggest a food-serving area. The gathered crowd indicates a social gathering, typical of a motorcycle gang meetup spot. The greenery indicates a green space, which is often used for public events. Therefore, the correct answer is A) A motorcycle gang meetup spot.

**Phase II: Answer generation**

**Answer:** The answer is (C). BECAUSE: The input Rationale is incorrect! The correct one is as follows: The presence of motorcycles parked in a row on the pavement indicates a public area, and the green stalls positioned nearby suggest a food-serving area. The crowd gathered indicates a social gathering, typical of a motorcycle gang meetup spot. The greenery indicates a green space, often used for public events. Therefore, the correct answer is C) A park.

*Figure 11.* Some error correction cases of MIND$_{base}$ on the M$^3$CoT dataset. Red denotes incorrect reasoning, green denotes correct reasoning, and "#" denotes the Question ID. The proposed MIND reasoning framework demonstrates excellent error correction capabilities.

only produces high-quality rationales but also possesses active logic discrimination and correction, which is one of the key factors enabling its significant performance improvements over existing multimodal CoT methods.

## D. Limitations, Failure Cases and Potential Risks

For the limitations, on the one hand, training MIND requires offline data pre-construction from RAD. Although the process is automated, one-off, and has minimal impact on training speed, it incurs additional data preparation costs. On the other hand, current validation focuses on VQA-style benchmarks. For the failure cases, although MIND's explicit correction mechanism improves robustness, it may still suffer from perceptual bias and reasoning errors in scenarios requiring fine-grained visual-semantic understanding, which is also a common challenge in this field. For the potential risks, limited original rationale quality given in initial samples may affect performance.

