# OpenReview forum: "MIND: Multi-rationale INtegrated Discriminative Reasoning Framework for Multi-modal Large Models"
_ICML.cc/2026/Conference — ICML 2026 regular_

### Official Review · Reviewer_nEGn · 2026-03-13

**Soundness:** 3
**Presentation:** 4
**Significance:** 3
**Originality:** 3
**Overall Recommendation:** 4
**Confidence:** 4

**Summary:**

This article analyzes a notable aspect of multimodal reasoning systems: the reliance of existing MCoT approaches on single-rationale supervision, which often leads to rigid reasoning patterns, weak robustness to misleading cues, and limited self-correction ability. To address these limitations, the paper proposes MIND. , a multi-rationale integrated discriminative reasoning framework for MLLMs. It introduces: (i) RAD, a data construction paradigm that co-generates diverse positive rationales and semantically inverted negative rationales; (ii) P2CL, a two-stage training scheme that first learns from diverse positive rationales (P2CL-I) and then performs active logic discrimination and correction conditioned on positive/negative rationales (P2CL-II); and (iii) MCA, a multi-rationale contrastive alignment loss that mines hard positives/negatives to widen semantic margins. On ScienceQA, A-OKVQA (MC), and M³CoT, the method achieves state-of-the-art performance compared with existing multimodal reasoning methods.

**Compliance With Llm Reviewing Policy:**

Affirmed.

**Key Questions For Authors:**

1. **Algorithmic gain vs. data scaling.** Since RAD expands the training rationales by up to 1000×, could the authors better disentangle the contribution of the proposed learning framework from the effect of large-scale data augmentation itself? In particular, controlled comparisons at matched data scales would help clarify this point.

2. **Fairness of comparison and effective model capacity.**  The reported model sizes (223M / 738M) seem to refer only to the trainable T5 backbone, while the full system also depends on a frozen BLIP2-flan-t5-xxl vision encoder (12B). Could the authors clarify the effective system capacity, external model assistance? Such as MIND with ViT-large, or Multimodal CoT with BLIP2-flan-t5-xxl.

3. **Robustness, reproducibility, and compute cost.**  The paper does not report multiple seeds, variance/statistical significance, or detailed compute accounting. Could the authors provide these results to better establish the robustness of the gains and the practical reproducibility of the method?

4. **Scope of generalization and limitations.**  The evaluation is limited to multiple-choice VQA-style benchmarks. Could the authors discuss whether MIND generalizes to broader multimodal reasoning settings (e.g., open-ended QA or more realistic applications), and provide a clearer discussion of limitations, failure cases, and potential risks?

**Limitations:**

1. The authors could discuss risks associated with **automatically generated rationales**, such as the propagation of biases or systematic errors from the large models used to generate positive and negative rationales.

2. The paper should acknowledge **computational cost,  and accessibility issues** for researchers with limited resources.

3. Finally, a clearer discussion of **methodological limitations** (e.g. evaluation limited to multiple-choice VQA benchmarks) would strengthen the transparency of the work.

**Strengths And Weaknesses:**

##### Strengths

###### Soundness

The paper proposes a reasonably well-structured framework consisting of RAD, P2CL, and MCA, with clearly defined training objectives and workflow. The empirical evaluation is relatively comprehensive, covering three benchmarks (ScienceQA, A-OKVQA, and M3CoT). Moreover, ablation studies, hyperparameter analysis, and qualitative examples help support the claimed improvements and clarify the contribution of each component.

###### Presentation

The overall presentation is clear and easy to follow, especially the cognitive-inspired paradigm of “Understand → Rethink → Correct”.  The framework is illustrated with helpful diagrams and qualitative examples comparing with prior methods, which improves readability and interpretability.

###### Significance

The paper addresses an important issue in multimodal reasoning: current models often rely on single rationales and lack the ability to detect misleading reasoning or correct logical errors.

###### Originality

The work provides a meaningful combination of several ideas: multi-rationale data construction, progressive correction learning, and contrastive alignment in rationale space. Specifically, the emphasis on explicit reasoning discrimination and correction represents an interesting perspective compared with standard single-rationale supervision.



##### Weaknesses



###### Soundness

The RAD expansion scales to up to 1000× rationales; it is hard to **disentangle algorithmic gains** from sheer data augmentation. Statistical significance, multiple seeds, and compute accounting are missing. In addition, **fairness of comparison is unclear**: the 223M/738M counts appear to exclude a frozen BLIP2-flan-t5-xxl vision encoder (12B) and strong Qwen2.5-VL-72B captions at inference. The effective capacity and external assistance may not be comparable to some baselines (Multimodal-CoT uses the ViT-large (~300M)).

###### Presentation

The paper provides **limited discussion** of limitations, failure cases, or potential risks of the approach.

###### Significance

Experiments focus primarily on VQA-style (Multiple-choice VQA) multi-modal reasoning benchmarks. It remains unclear whether the framework generalizes well to broader multimodal reasoning tasks or real-world applications. Although the performance improvements are strong on the evaluated datasets, the broader impact on **general** multimodal reasoning systems still needs further validation.

###### Originality

None.

---

> ### Author Rebuttal · Authors · 2026-03-30
>
> **We sincerely thank Reviewer nEGn for the constructive comments and appreciation of our strengths**, such as interesting perspective, methodological novelty, clear presentation, and superior performance. The reviewer raised several insightful questions, which we address in detail below. (**W: Weaknesses, L: Limitations**)
> > **Q1&W1.** Can the authors better disentangle the gains from the learning framework and those from RAD’s data scaling, e.g., via matched-scale comparisons?
>
> **R1.** Thanks for your valuable question. We agree that it's necessary to distinguish between the gains of data scaling and framework design. Since RAD and P2CL/MCA are jointly design, we compared them under matched settings. Specifically:
> - **Under the same learning framework, data scaling is beneficial, but the gains are gradual and saturating.** From Table 6 in our paper, expanding rationales from the original scale to ×10/×50/×100/×500/×1000 improves accuracy on ScienceQA from 90.29 to 91.42/91.56/91.58/91.91/92.29.
> - **At matched data scale, the learning framework still brings clear gains.** From Table 9 in our paper, compared with MCA alone, adding P2CL brings a further gain of 1.69%-2.54%; compared with P2CL alone, adding MCA still yields an additional 0.14%-1.04%.
>
> We will present a detailed description in the revision.
> > **Q2&W1.** Can the authors clarify the effective system capacity and external assistance, and provide fairness comparison with matching settings?
>
> **R2.** Thanks for the important reminder.
> - The reported “223M/738M” denote only the trainable T5 backbone and do not include the external modules (BLIP2-flan-t5-xxl visual encoder and Qwen2.5-VL-72B caption generator).
> - We further performed **a stricter comparison with matching settings.** From Table R-4, $MIND_{base}$ still outperforms $Multimodal\text{-}CoT_{base}$ by 3.82%-4.50%. It suggests that the gains are not solely from stronger external modules, but also from MIND itself.
>
> ***Table R-4. Comparison under the same visual encoder (former) and caption generator (latter) on ScienceQA.***
> | Schemes | BLIP2-flan-t5-xxl + InstructBLIP-7B | ViT-large-patch32-384 + Qwen2.5-VL-72B | BLIP2-flan-t5-xxl + Qwen2.5-VL-72B |
> |---|---:|---:|---:|
> | $Multimodal\text{-}CoT_{base}$ | 86.98 (+3.82) | 87.53 (+4.24) | 87.79 (+4.50) |
> | $MIND_{base}$ | 90.80 | 91.77 | 92.29 |
>
> We will add these details in the revision.
> > **Q3&W1&L2.** Can the authors report multiple seeds, variance significance, and detailed compute cost to better demonstrate the robustness and reproducibility of the method?
>
> **R3.** Thanks for the important reminder.
> - We used a fixed seed of 42 in all experiments, **ensuring reproducibility**. Meanwhile, $MIND_{base}$ is relatively stable on M³CoT over five seeds (mean 57.40, variance 0.35).
> - For **compute cost**, since MIND randomly samples from the constructed rationale pool during training, it does not increase linearly with the total rationales. On ScienceQA, the training time per epoch is 0.14h / 0.17h for $Multimodal\text{-}CoT_{base}$ / $MIND_{base}$ on a single H20 GPU, and 0.29h / 0.31h for $Multimodal\text{-}CoT_{large}$ / $MIND_{large}$ on four H20 GPUs, showing that MIND introduces only minor additional training overhead.
> - **We will release the code and data to support reproducibility and future research.**
>
> We will add these details in the revision.
> > **Q4&W2&W3&L3.** Could the authors discuss whether MIND has broader applicability beyond multiple-choice VQA, and clarify its limitations, failure cases, and potential risks?
>
> **R4.** Thanks for the constructive suggestion. MIND is not limited to VQA-style tasks. Specifically:
> - **MIND is a training framework rather than a task-specific design.**
> - Current experiments focus on VQA-style benchmarks, mainly for **controllable and quantifiable verification**.
> - MIND learns **reasoning correction rather than simple answer mapping.** From the cases in Fig. 9-11 of our paper, it can identify reasoning deviations and reconstruct correct logic from erroneous rationales, showing its potential for more open-ended scenarios.
> - We are also continuously expanding MIND. **From Table R-3 in the response to Reviewer xB8J, we extend MIND to Qwen2.5-VL-7B, Qwen3-VL-8B, and Qwen3.5-9B.** Compared to MIND without MCA and P2CL, the final MIND showed significant gains of **8.54%-12.68%**.
> - Regarding limitations, failure cases, and potential risks:
>     - **Limitations**: The additional offline RAD construction cost and validation mainly on VQA-style benchmarks.
>     - **Failure cases**: Fine-grained visual-semantic reasoning, which is also a common challenge.
>     - **Potential risks**: Limited original rationale quality given in initial samples may affect performance.
>
> We will add these explanations in the revision.
> > **L1.** The authors could discuss risks associated with automatically generated rationales.
>
> **R5.** Thanks for the insightful comment. Please refer to our response of **R1 of the Reviewer S3RQ.**

---

> > ### Author Rebuttal · Reviewer_nEGn · 2026-04-02
> >
> > The authors have addressed my concerns. I will maintain my positive score.

---

> > > ### Author Response · Authors · 2026-04-03
> > >
> > > Thank you for taking the time to review our responses. We truly appreciate your positive feedback and your support!

---

### Official Review · Reviewer_xB8J · 2026-03-13

**Soundness:** 3
**Presentation:** 3
**Significance:** 2
**Originality:** 3
**Overall Recommendation:** 4
**Confidence:** 3

**Summary:**

This paper proposes MIND, a multimodal reasoning framework designed to move MLLMs from passive imitation-based reasoning to active discriminative reasoning. The core idea is to train models on diverse positive and negative rationales rather than a single ground-truth rationale. The framework consists of three components: a RAD paradigm for automatically generating multi-rationale training data, a two-stage P2CL training strategy that first learns from diverse correct rationales and then learns to identify and correct erroneous ones, and an MCA contrastive alignment objective that separates correct and incorrect rationale representations in the embedding space. The system is evaluated on ScienceQA, A-OKVQA, and M³CoT, achieving state-of-the-art performance against Multimodal-CoT and related baselines.

**Compliance With Llm Reviewing Policy:**

Affirmed.

**Final Justification:**

After reading the rebuttal, I raised my score from weak reject to weak accept. The scalability evidence in Table R-3 was convincing and addressed my main concern about the framework's generalizability beyond small encoder-decoder models. The methodology is sound and the empirical results are strong across benchmarks. My remaining concern that the central claim of a reasoning paradigm shift is not fully supported by accuracy-based evaluation alone is not fully resolved, but I believe the paper's contributions are sufficient. I encourage the authors to include more process-level analysis.

**Key Questions For Authors:**

The model is trained to discriminate between positive and negative rationales, where negatives are generated by semantically inverting correct ones. How does the model perform when tested on rationale errors it was not trained on, for example, rationales that are plausible but factually wrong, or that contain irrelevant distractors rather than direct logical negations? Is the discrimination ability specific to the training distribution of negatives?

The paper's core motivation is that single-rationale supervision leads to shallow reasoning. But the evaluation only measures final answer accuracy. Is there any analysis showing that MIND's reasoning process is qualitatively different, for instance, that it makes different errors, is more robust to input perturbations, or generalizes better to out-of-distribution tasks?

The framework is evaluated on relatively small encoder-decoder models. Would the same training paradigm, particularly the P2CL strategy and negative rationale exposure, provide meaningful gains when applied to larger and more capable base models? Is there reason to believe the approach scales?

**Strengths And Weaknesses:**

Strengths:

The paper addresses an underappreciated limitation of existing multimodal chain-of-thought methods: that training on a single standard rationale encourages surface-level pattern matching rather than robust logical reasoning. The proposed solution, exposing the model to both diverse correct rationales and deliberately constructed incorrect ones, is intuitive and well-motivated by the cognitive framing of "Understand → Rethink → Correct." The RAD data construction pipeline is practical and scalable, leveraging existing large models to automatically generate the positive and negative rationale pools. The two-stage training design is coherent, with the first phase building semantic understanding and the second phase developing active error correction. The error-correction qualitative examples in the appendix are illustrative and support the paper's claims. Results are consistently strong across all three benchmarks.

Weaknesses:

The evaluation does not test whether the model has actually learned to reason or just learned to correct. The paper's central claim is that MIND moves models from passive imitation to active discrimination. However, the evaluation is entirely accuracy-based; the model is measured on whether it produces the right answer, not on whether its reasoning process is actually more robust or generalizable. There is no analysis of what happens when the model is tested on rationales that fall outside the distribution of negatives it was trained on, or whether the discrimination ability transfers to genuinely novel reasoning challenges. Without this, it is unclear whether the model has internalized logical discrimination or has simply learned a new form of pattern matching, this time over positive/negative rationale pairs rather than single rationales.

The negative rationale construction strategy is too narrowly defined to support the paper's broader claims. The negative rationales are generated by prompting models to semantically invert the original correct rationale while preserving structure. This produces a specific and constrained type of error, surface-level logical negation. Real reasoning failures in the wild are far more diverse: they include incomplete causal chains, irrelevant evidence, plausible-sounding but wrong conclusions, and so on. The paper claims MIND endows models with human-like self-correction capabilities, but the training data only covers one narrow failure mode. Whether the learned discrimination generalizes to other error types is never tested.

The base model is substantially weaker than the current state of the art, which limits the significance of the results. MIND is built on a T5-based encoder-decoder architecture with hundreds of millions of parameters. The comparisons in the paper are largely against similarly small models from several years ago. The paper does compare against some larger zero-shot models but attributes MIND's advantage to its training paradigm. It is unclear whether the proposed framework would provide meaningful gains when applied to more capable modern base models, and the lack of any such experiment leaves the broader impact of the approach uncertain.

---

> ### Author Rebuttal · Authors · 2026-03-30
>
> **We sincerely thank Reviewer xB8J for the constructive comments and appreciation of our strengths**, such as well-motivated, methodological novelty, and superior performance. The reviewer raised several insightful questions, which we address in detail below. (**W: Weaknesses**)
> > **Q1&W2.** How well does MIND generalize to unseen types of rationale errors beyond the semantically inverted negatives used in training? Is its discrimination ability specific to the training distribution of negatives?
>
> **R1.** Thanks for your insightful question. **MIND’s discriminative ability is not limited to the training negative distribution**. It learns to identify and correct errors on hard negative rationales that are similar but semantically conflicting, which gives it generalization ability to unseen rationale errors. Specifically:
> - **The training negative rationales are not templated noise, but task-aware hard negatives.** Thus, the model learns not simple pattern matching, but semantic conflict boundaries.
> - **We further construct five unseen rationale error types** using Qwen3-235B-22A besides the original error type. From Table R-2, MIND's results on other five error types are generally close to “Original”, indicating good generalization to unseen rationale errors.
> - Moreover, **P2CL-II learns logical discrimination and correction, not negative sample classification.** This is supported by Table 8 in our paper: “Neg→Pos” (91.72) is significantly better than “Neg→N/A” (90.64), and “Pos→Pos or Neg→Pos” (92.29) is also clearly better than “Pos→Pos or Neg→N/A” (91.39).
>
> ***Table R-2. MIND performance across rationale error types (100 test samples/dataset).***
> | Error Type | Original | Plausible but Wrong | Irrelevant Distractor | Incomplete Reasoning | Selective Evidence | Logical Misbinding |
> |---|---:|---:|---:|---:|---:|---:|
> | ScienceQA | 90.0% | 91.0% | 88.0% | 93.0% | 89.0% | 91.0% |
> | A-OKVQA | 67.0% | 68.0% | 67.0% | 67.0% | 65.0% | 69.0% |
> | M³CoT | 59.0% | 55.0% | 59.0% | 60.0% | 56.0% | 60.0% |
>
> We will add the experiments in the revision.
> > **Q2&W1.** Beyond final answer accuracy, is there evidence that MIND learns deeper or qualitatively different reasoning?
>
> **R2.** Thanks for your important question. We agree that accuracy alone is insufficient. Therefore, we want to emphasize that our evidence goes beyond accuracy and comes from training objectives, ablation results, qualitative cases, and error perturbation testing. Specifically:
> - From the **training objective**, MIND does not simply learn answer mapping, but rather learns the correct reasoning logic (P2CL-I) and explicitly identifies and corrects reasoning biases (P2CL-II).
> - From the **ablation results** in Table 8 of our paper, “Pos→Pos or Neg→ Pos” (92.29%) is significantly better than “Pos→Pos or Neg→N/A” (91.39%). This shows that its main gain comes from active logical discrimination and error correction, rather than simply stronger answer prediction.
> - From the **qualitative cases in Figs. 6-11** of our paper, the improvement of MIND is also reflected in the reasoning process itself.
> - From the **perturbation test in Table R-2**, the model’s discrimination and correction ability is not limited to a specific negative pattern, but shows robustness to broader reasoning deviations.
>
> We will present a detailed description in the revision.
>
> > **Q3&W3.** Does the proposed training paradigm remain effective when applied to larger and more capable base models?
>
> **R3.** Thanks for your valuable question. MIND is not merely a compensation for small models, but a paradigm-level training enhancement that remains effective at larger scales. Specifically:
> - **MIND is not tied to a specific foundation model.** P2CL explicitly models the "Understand→Rethink→Correct" process, while MCA further strengthens the semantic discriminative boundaries. MIND focuses on the training paradigm itself and has excellent scalability.
> - **The results in our paper show consistent gains across different model scales.** On M³CoT, $MIND_{base}$ and $MIND_{large}$ outperform their corresponding Multimodal-CoT baselines by 12.53% and 12.83%.
> - **Notably, we are continuously exploring the extensions of MIND, and have extended MIND to Qwen2.5-VL, Qwen3-VL, and Qwen3.5.** To date, we have completed comparative experiments on M³CoT, which includes multi-domain problems. From Table R-3, compared to the original model without SFT, MIND improves by **18.63%-24.89%**. Compared to the MIND without MCA and P2CL, it still improves over **8.54%-12.68%.**
>
> ***Table R-3. Performance of MIND extended to different Qwen models on M³CoT.***
> | Schemes | Qwen2.5-VL-7B | Qwen3-VL-8B | Qwen3.5-9B |
> |---|---:|---:|---:|
> | Original (No SFT) | 60.96 (+24.89) | 65.83 (+22.31) | 70.02 (+18.63) |
> | Baseline (MIND w/o MCA, P2CL) | 77.31 (+8.54) | 77.74 (+10.40) | 75.97 (+12.68) |
> | MIND | **85.85** | **88.14** | **88.65** |
>
> We will add the experiments and analyses in the revision.

---

> > ### Author Rebuttal · Reviewer_xB8J · 2026-04-03
> >
> > Thank you for the detailed reply. I appreciate the additional experiments provided, particularly Table R-3 demonstrating MIND's effectiveness on larger Qwen-series models. This addresses my concern about scalability (W3), and I am satisfied on that point.
> >
> > My core remaining concern is W1: the paper's central claim is a paradigm shift from passive imitation to active discriminative reasoning, but the evaluation remains entirely accuracy-based. The rebuttal points to ablation results and qualitative figures, but these do not directly demonstrate that the reasoning process is qualitatively more robust; they show that components contribute to accuracy on in-distribution benchmarks, which is a different claim.
> >
> > My follow-up questions for the authors are: (1) Is there any evaluation on out-of-distribution tasks or input perturbation experiments that could more directly support the claim of deeper reasoning? (2) For naturally occurring errors that appear in the wild, do you have any analysis of whether MIND fails differently from Multimodal-CoT?

---

> > > ### Author Response · Authors · 2026-04-06
> > >
> > > **We sincerely thank Reviewer xB8J for taking the time to review our first round of responses and for the positive recognition (Score: 3 → 4).** Meanwhile, we address the follow-up questions in detail below.
> > > > **Q1:** Is there any evaluation on out-of-distribution tasks or input perturbation experiments that could more directly support the claim of deeper reasoning?
> > >
> > > **R1:** Thanks for your important question.
> > > - **Evaluation on out-of-distribution task.** We used M³CoT for training and ScienceQA for testing. From Table R-5, compared to the original model without SFT, the "Baseline" hurts cross-dataset generalization, whereas our MIND still achieves gains of 1.18%-6.32%. Furthermore, MIND outperforms the "Baseline" by 3.59%-7.10%. These results indicate that MIND has stronger generalization ability and deeper reasoning in cross-dataset scenarios.
> > > - **We further extend the input perturbation evaluation in Table R-2 and report the results in Table R-6.** "Normal" denotes normal testing without perturbation. Compared to "Normal", Multimodal-CoT drops sharply under all six error perturbations, including the original error type used in RAD to generate negative samples, while MIND remains stable and even improves slightly on M³CoT. This indicates that MIND is more robust to both seen and unseen reasoning biases and more directly supports the effectiveness of the active logic discrimination and correction mechanism in P2CL-II. This further highlights its potential for real-world applications.
> > >
> > > ***Table R-5. Evaluation on the out-of-distribution task. Train: M³CoT. Test: ScienceQA.***
> > > |Schemes|Qwen2.5-VL-7B|Qwen3-VL-8B|Qwen3.5-9B|
> > > |---|---:|---:|---:|
> > > |Origin (No SFT)|82.74 (+6.32)|90.57 (+1.18)|90.71 (+1.39)|
> > > |Baseline (MIND w/o MCA, P2CL)|83.80 (+5.26)|88.16 (+3.59)|85.00 (+7.10)|
> > > |MIND|**89.06**|**91.75**|**92.10**|
> > >
> > > ***Table R-6. Evaluation on input perturbation task (100 test samples/dataset). The base models are all T5-base.***
> > > |Methods|Dataset|Normal|Original Error Type|Plausible but Wrong|Irrelevant Distractor|Incomplete Reasoning|Selective Evidence|Logical Misbinding|
> > > |---|---|---:|---:|---:|---:|---:|---:|---:|
> > > |Multimodal-CoT|ScienceQA|84.0%|34.0%|43.0%|47.0%|66.0%|35.0%|41.0%|
> > > |MIND|ScienceQA|91.0%|90.0%|91.0%|88.0%|93.0%|89.0%|91.0%|
> > > |Multimodal-CoT|A-OKVQA|52.0%|16.0%|19.0%|16.0%|37.0%|4.0%|19.0%|
> > > |MIND|A-OKVQA|68.0%|67.0%|68.0%|67.0%|67.0%|65.0%|69.0%|
> > > |Multimodal-CoT|M³CoT|42.0%|25.0%|34.0%|14.0%|46.0%|8.0%|13.0%|
> > > |MIND|M³CoT|55.0%|59.0%|55.0%|59.0%|60.0%|56.0%|60.0%|
> > >
> > > We will add the experiments in the revision.
> > >
> > > > **Q2:** For naturally occurring errors that appear in the wild, do you have any analysis of whether MIND fails differently from Multimodal-CoT?
> > >
> > > **R2:** Thanks for your insightful question.
> > > - From Table R-6, under various simulated natural errors, Multimodal-CoT degrades sharply, while MIND remains generally stable. **This matches their mechanism difference.** Multimodal-CoT lacks explicit error discrimination and tends to follow misleading cues, whereas MIND explicitly models "Understand → Rethink → Correct", thereby ensuring the correctness of both reasoning and answer.
> > > - Based on our observations of fail cases across multiple datasets, Multimodal-CoT failures are related to misleading clues, logical errors, and insufficient evidence. **In contrast, MIND effectively mitigates these failures**, but it can still fail when fine-grained visual-semantic clues are required, which is also a common challenge.
> > > - **We further conduct a quantitative evaluation of the rationale quality.** To avoid subjective human factors, we used a larger model (Qwen3-VL-30B-A3B) to score. Since MIND explicitly models discrimination and correction by P2CL-II, we report both MIND (P2CL-I) and MIND (P2CL-II). From Table R-7, compared with Multimodal-CoT, MIND (P2CL-I) outperforms on all four dimensions with an overall gain of 0.28. Furthermore, MIND (P2CL-II) further improves across all dimensions. This shows that MIND improves not only final answer accuracy, but also rationale quality and the model’s response to erroneous reasoning cues.
> > > - **This trend is consistent across other datasets.** Our evaluations on A-OKVQA and M³CoT also show that the "Overall" scores for Multimodal-CoT/MIND(P2CL-I)/MIND(P2CL-II) are 6.85/7.47/7.94 and 3.63/5.34/5.54. These further validate effectiveness and robustness of the MIND.
> > >
> > > ***Table R-7. Rationale quality evaluation on ScienceQA. The base models are all T5-base. Total score: 10.***
> > > |Rationale output|Correctness|Relevance|Coherence|Solving process|Overall (Mean)|
> > > |---|---:|---:|---:|---:|---:|
> > > |Multimodal-CoT|7.49|8.64|8.35|8.00|8.12|
> > > |MIND (P2CL-I)|8.01|8.73|8.55|8.32|8.40|
> > > |MIND (P2CL-II)|**8.08**|**8.88**|**8.71**|**8.44**|**8.53**|
> > >
> > > We will add the experiments and analyses in the revision.
> > >
> > > ---
> > > **We sincerely hope that the above responses can address your concerns and obtain your further recognition.** Thank you again for taking the time to review our responses.

---

### Official Review · Reviewer_S3RQ · 2026-03-22

**Soundness:** 3
**Presentation:** 4
**Significance:** 3
**Originality:** 3
**Overall Recommendation:** 5
**Confidence:** 4

**Summary:**

The paper proposes MIND, a Multi-rationale INtegrated Discriminative reasoning framework designed to enhance the reasoning capabilities of Multimodal Large Language Models (MLLMs). The authors identify that current MLLMs suffer from rigid reasoning patterns and susceptibility to misleading cues due to their reliance on single-rationale supervision. To address this problem, the authors aim to endow MLLMs with a human-like "Understand -> Rethink -> Correct" cognitive process.The framework is built on three core components. (1) The Rationale Augmentation and Discrimination (RAD) paradigm leverages existing large models to automatically generate diverse positive rationales and challenging negative rationales; (2) Progressive Two-stage Correction Learning (P2CL) strategy utilizes these rationales: Phase I focuses on multi-rationale positive learning, and Phase II trains the model to actively discriminate and correct logical errors using positive-negative rationale pairs; (3) a Multi-rationale Contrastive Alignment (MCA) optimization strategy is introduced to explicitly separate positive and negative rationales in the embedding space. The authors validate their approach on the ScienceQA, A-OKVQA, and M3CoT datasets and demonstrate state-of-the-art performance.

**Compliance With Llm Reviewing Policy:**

Affirmed.

**Key Questions For Authors:**

Q1. Regarding the RAD paradigm, how sensitive is the MIND framework to the hallucination rate or reasoning flaws of the specific teacher models used to generate the positive and negative rationales? And how much efforts should be paid to handle these rationales.

Q2. In the MCA optimization, the authors utilize a margin-based contrastive loss with a margin m=0.2. Did the authors conduct more experiments with other contrastive formulations (such as InfoNCE)?

Q3. The P2CL-II phase uses negative rationales to explicitly trigger the "Correct" behavior. During inference, does the model ever "over-correct" or second-guess itself when presented with a slightly ambiguous, like "wait"?

**Strengths And Weaknesses:**

S1. Data Contribution: The Rationale Augmentation and Discrimination (RAD) paradigm systematically generates diverse positive and negative rationales. By expanding dataset rationales by up to 1000x and promising to open-source the data and code , the authors provide a highly extensible and unified data foundation for the community.

S2. Performance: The proposed framework achieves clear State-of-the-Art (SOTA) results. Notably, the $MIND_{base}$ model outperforms the strong Multimodal-CoT baseline by a massive 20.0% on the A-OKVQA dataset , and by 6.98% on the ScienceQA dataset, demonstrating high practical utility and effectiveness.

S3. Evaluation: The authors exhibit a highly serious and commendable attitude toward empirical validation. They evaluate MIND comprehensively across three distinct datasets (ScienceQA, A-OKVQA, and M³CoT) and provide remarkably thorough ablation studies. This includes break-down analysis , detailed investigations of rationale quality and quantity , and hyperparameter sensitivity for the MCA strategy.

W1. Novelty: While the combination of techniques is highly effective, the individual components represent minor algorithmic contributions. The P2CL strategy is fundamentally a two-stage supervised finetuning approach utilizing maximum likelihood generation , and the MCA strategy relies on a standard margin-based contrastive loss.

W2. Effectiveness: The ablation studies reveal that the Multi-rationale Contrastive Alignment (MCA) strategy offers very limited benefits on its own. For example, it only yields a 0.07% accuracy improvement on the ScienceQA dataset when applied without the P2CL strategy.

W3. Distillation: The framework's success is deeply tied to the quality of the rationales generated by massive models like Qwen3-235B-A22 and DeepSeek-V3. This dependency suggests the framework essentially distills reasoning capabilities from these massive models, which slightly diminishes the core novelty of the reasoning architecture itself.

---

> ### Author Rebuttal · Authors · 2026-03-30
>
> **We sincerely thank Reviewer S3RQ for the constructive comments and appreciation of our strengths**, such as data contribution, full experiments, in-depth analysis, and superior performance. The reviewer raised several insightful questions, which we address in detail below. (**W: Weaknesses**)
> > **Q1&W3.** How sensitive is MIND to hallucinations or reasoning flaws in the teacher-generated rationales used by RAD, and how much effort is required to handle these rationales?
>
> **R1.** Thanks for the insightful question.
> - We agree that MIND is not entirely insensitive to teacher rationale quality. However, MIND does not directly distill teacher outputs. Instead, it reduces reliance on any single teacher through RAD and P2CL. Therefore, MIND remains robust to the teacher noise. Specifically:
>     - **Different rationale generators affect performance, but the overall trend remains stable**. RAD’s multi-source construction further reduces this sensitivity and yields gains (Table 5 in our paper).
>     - **RAD further reduces sensitivity via controlled generation and filtering**. Positive rationales preserve semantics, while negative ones use minor yet semantically reversed edits. Meanwhile, task information is incorporated into the prompt, and outputs are filtered to ensure semantics and valid format.
>     - **MIND does not simply imitate teacher outputs**. P2CL-I learns shared logic across multiple positive rationales, while P2CL-II further uses positive/negative rationales to guide the model in identifying and correcting erroneous reasoning.
> - The effort/cost is mainly **one-time offline automated preprocessing**, not intensive manual annotation or cleaning. RAD also uses batch-wise generation for efficiency, and training only samples from pre-built rationale pools, the extra online overhead is minimal.
>
> We will clarify this in the revision.
> > **Q2.** Did the authors compare the loss in MCA with other contrastive formulations (InfoNCE)?
>
> **R2.** Thanks for the important question. From Table R-1, InfoNCE with different *T* consistently performs worse than the margin-based contrastive loss. We believe this is because the latter is better aligned with MCA’s objective of hard positive aggregation and hard negative separation, i.e., directly shaping the local discrimination boundary, whereas InfoNCE is more inclined toward global normalized competition. We will add the experiments in the revision.
>
> ***Table R-1. Comparison of different losses. T: temperature.***
>
> | Loss setting | ScienceQA | A-OKVQA | M³CoT |
> |---|---:|---:|---:|
> | InfoNCE (T=0.05) | 91.89 | 69.52 | 56.08 |
> | InfoNCE (T=0.07) | 92.03 | 70.13 | 56.56 |
> | InfoNCE (T=0.1) | 92.05 | 69.87 | 56.34 |
> | Margin-based Contrastive | **92.29** | **70.57** | **57.38** |
>
> >**Q3.** During inference, does P2CL-II make the model prone to over-correct or second-guess when presented with a slightly ambiguous?
>
> **R3.** Thanks for the valuable question. Slight over-correction can occur in a few cases, but it mainly reflects a conservative triggering of the correction template rather than true logical instability. The final rationale is usually still correct in such cases. Specifically:
> - **P2CL-II is a conditional discrimination and correction mechanism.** It emphasizes stability for positive rationales and corrects negative ones, helping avoid indiscriminate second-guessing.
> - **The overall effect of P2CL-II is significant.** From Table 8 in our paper, removing it reduces the accuracy from 92.29% to 90.36%. Meanwhile, "Pos→Pos or Neg→Pos" significantly outperforms "Pos→Pos or Neg→N/A", indicating that **P2CL-II mainly improves logic discrimination and correction rather than causing over-correction.** The qualitative cases in Figs. 9-11 of our paper further support this.
>
> We will add relevant discussion in the revision.
> >**W1&W2.** The value and effectiveness of MIND need be clarified more clearly. Especially, MCA alone brings only a marginal gain on ScienceQA.
>
> **R4.** Thanks for the valuable comment. **MIND’s core contribution is not only in data, but also in enabling MLLMs to move from passive imitation to active discrimination and correction.** It boasts strong performance and scalability. Specifically:
> - P2CL learns shared reasoning logic and explicitly trains the model to identify and correct erroneous reasoning, enabling “Understand→Rethink→Correct.”, while MCA further strengthens the semantic discriminative boundaries.
> - **Both P2CL and MCA bring effective gains.** The small gain of "+0.07%" on ScienceQA mainly reflects its already strong baseline. Across all datasets (Table 9 in our paper), MCA alone yields gains of 0.07%-3.02%, P2CL alone brings gains of 1.86%-4.28%.
> - Notably, **we further extend MIND to Qwen2.5-VL-7B, Qwen3-VL-8B, and Qwen3.5-9B.** From **Table R-3 in the response to Reviewer xB8J**, compared to MIND without MCA and P2CL, the final MIND showed a significant improvement of **8.54%-12.68%**.
>
> We will provide a clearer description in the revision.

---

> > ### Author Rebuttal · Reviewer_S3RQ · 2026-04-03
> >
> > Thank the authors for the comprehensive and thoughtful rebuttal. The additional experiments and detailed clarifications have effectively addressed my initial concerns. Here are my specific thoughts on the provided responses:
> >
> > - **Regarding Q1 & W3 (Teacher Dependency and Hallucinations):** The explanation of how the RAD paradigm mitigates reliance on a single teacher model through multi-source construction is convincing. It clarifies that the framework is not merely doing direct knowledge distillation, but rather actively shaping a robust reasoning space.
> > - **Regarding Q2 (Contrastive Loss Formulation):** I appreciate the inclusion of Table R-1. The empirical comparison clearly demonstrates that the margin-based contrastive loss outperforms InfoNCE across the tested temperatures for this specific task.
> > - **Regarding Q3 (Over-correction):** The clarification that P2CL-II operates conditionally to emphasize stability for positive rationales is helpful.
> > - **Regarding W1 & W2 (Novelty):** The new scaling results applying the MIND framework to larger, modern architectures like Qwen2.5-VL-7B, Qwen3-VL-8B, and Qwen3.5-9B are excellent additions. Table R3 addresses my concern about the marginal gains seen in some isolated ablations and proves the framework's broader scalability and value.
> >
> > Overall, the rebuttal strengthens my confidence in the submission's contributions. I maintain my recommendation to support this paper.

---

> > > ### Author Response · Authors · 2026-04-03
> > >
> > > Thank you for taking the time to review our responses. We truly appreciate your positive feedback and your support!

---

### Decision · Program_Chairs · 2026-04-30

**Decision:**

Accept (regular)

**Comment:**

The paper proposes MIND, a Multi-rationale INtegrated Discriminative reasoning framework designed to endow multimodal large language models (MLLMs) with a human-like "Understand → Rethink → Correct" cognitive process through three components: a Rationale Augmentation and Discrimination (RAD) data paradigm, a Progressive Two-stage Correction Learning (P2CL) strategy, and a Multi-rationale Contrastive Alignment (MCA) optimization. Its main contribution lies in shifting MLLM reasoning from passive imitation of single-rationale supervision to active discriminative reasoning over diverse positive and deliberately constructed negative rationales, supported by an extensible open-source data pipeline.

**Strengths:**
- The cognitive-inspired framing of "Understand → Rethink → Correct" is intuitive and well-motivated, addressing an underappreciated limitation of single-rationale supervision in existing multimodal chain-of-thought methods (noted by xB8J and nEGn).
- The RAD data construction pipeline is practical and scalable, expanding rationales by up to 1000× and providing a unified, extensible data foundation that the authors commit to open-source (S3RQ, nEGn).
- Empirical evaluation is thorough and achieves consistent SOTA results across ScienceQA, A-OKVQA, and M³CoT, with comprehensive ablations, hyperparameter sensitivity analysis, and qualitative examples (S3RQ, nEGn).

**Weaknesses:**
- Limited algorithmic novelty of individual components — P2CL is essentially two-stage supervised finetuning and MCA uses a standard margin-based contrastive loss (S3RQ).
- The central claim of a "paradigm shift" from imitation to discriminative reasoning is supported only by accuracy-based evaluation, without direct process-level analysis of reasoning robustness or generalization to naturally occurring error types (xB8J).
- Narrowly defined negative rationales (semantic inversion) and concerns about whether discrimination ability generalizes to broader real-world error modes such as plausible-but-wrong or irrelevant distractors (xB8J).
- Fairness of comparison is unclear due to external modules (BLIP2-flan-t5-xxl encoder, Qwen2.5-VL-72B captions) not reflected in reported parameter counts, and the base models are weaker than current SOTA, raising concerns about scalability and gains from data scaling vs. framework design (nEGn, xB8J).

The reviewers reached a positive consensus post-rebuttal— Reviewer nEGn's concerns were fully resolved, Reviewer S3RQ maintained support, and Reviewer xB8J raised their score from weak reject to weak accept after the authors provided convincing scalability results on larger Qwen-series models, out-of-distribution evaluation, and perturbation robustness experiments. While the paper's claim of deeper reasoning is not fully substantiated beyond accuracy metrics, the strong empirical gains, comprehensive rebuttal evidence, and practical data contribution justify acceptance.